# Improved ozone monitoring by ground-based FTIR spectrometry

Omaira Elena García[1], Esther Sanromá[1, a], Matthias Schneider[2], Frank Hase[2], Sergio
Fabián León-Luis[1, b], Thomas Blumenstock[2], Eliezer Sepúlveda[1], Alberto Redondas[1], Virgilio Carreño[1],
Carlos Torres[1], and Natalia Prats[1]

[1]Izaña Atmospheric Research Centre (IARC), State Meteorological Agency of Spain (AEMet), Santa Cruz de Tenerife, Spain.
[2]Karlsruhe Institute of Technology (KIT), Institute of Meteorology and Climate Research (IMK-ASF), Karlsruhe, Germany.
[a]Now at: Employment Observatory of the Canary Islands (OBECAN), Santa Cruz de Tenerife, Spain.
[b]Now at: TRAGSATEC, Madrid, Spain.

**Correspondence:** ogarciar@aemet.es

**Abstract.** Accurate observations of atmospheric ozone ($O_3$) are essential to monitor in detail its key role in atmospheric chemistry. The present paper examines the performance of different $O_3$ retrieval strategies from FTIR (Fourier Transform InfraRed) spectrometry by using the 20-year time series of the high-resolution solar spectra acquired from 1999 to 2018 at the subtropical Izaña Observatory (IZO, Spain) within NDACC (Network for the Detection of Atmospheric Composition Change).
In particular, the effect of two of the most influential factors has been investigated: the inclusion of a simultaneous atmospheric temperature profile fit and spectral $O_3$ absorption lines used for the retrievals (the broad spectral region of 1000-1005 $cm^{-1}$ and single micro-windows between 991-1014 $cm^{-1}$). Additionally, the water vapour ($H_2O$) interference on $O_3$ retrievals has been evaluated, with the aim of providing an improved $O_3$ strategy that minimises its impact and, therefore, could be applied at any NDACC FTIR station under different humidity conditions. The theoretical and experimental quality assessments of the
different FTIR $O_3$ products (total column -TC- amounts and volume mixing ratio -VMR- profiles) provide consistent results. Combining a simultaneous temperature retrieval and an optimal selection of single $O_3$ micro-windows results in superior FTIR $O_3$ products, with a precision better than 0.6-0.7% for $O_3$ TCs as compared to coincident NDACC Brewer observations taken as reference. However, this improvement can only be achieved provided the FTIR spectrometer is properly characterised and stable over time. For unstable instruments, the temperature fit has been found to exhibit a strong negative influence on $O_3$
retrievals due to the increase in the cross-interference between the temperature retrieval and instrumental performance (given by the instrumental line shape function and measurement noise), which leads to the precision of FTIR $O_3$ TCs worsening up to 2%. This cross-interference becomes especially noticeable beyond the upper troposphere/lower stratosphere as documented theoretically as well as experimentally by comparing FTIR $O_3$ profiles to those measured using Electrochemical Concentration Cell (ECC) sondes within NDACC. Consequently, it should be taken into account for the reliable monitoring of $O_3$ vertical
distribution, especially in long-term timescales.

## 1   Introduction

Monitoring atmospheric composition is crucial for understanding present climate and foreseeing possible future changes and is, therefore, the basis for the design and implementation of efficient climate-change mitigation and adaptation policies. Among the

atmospheric gases with important climate effects, ozone ($O_3$) plays a vital role in atmospheric chemistry. In the stratosphere it
absorbs a large part of the biologically damaging ultraviolet sunlight, allowing only a small amount to reach the Earth's surface.
Likewise, absorption of the ultraviolet radiation heats, stratifies, and determines the vertical stability of the middle atmosphere.
In the troposphere, $O_3$ absorbs infrared radiation, acting as an important greenhouse gas, affects the oxidising capacity of the
atmosphere, and is a harmful phytotoxicant to public health (Cuevas et al., 2013; WMO, 2014a, 2018; Gaudel et al., 2018, and
references therein).

Stratospheric $O_3$ abundances have shown a significant decrease in global levels from the 1980s to the 1990s, mainly attributed
to the increase of anthropogenic emission of Ozone Depleting Substances (ODSs) during that period (WMO, 2014a, 2018).
The implementation of the 1987 Montreal Protocol and its amendments and adjustments has stopped global $O_3$ decay by
controlling the ODSs emissions, with $O_3$ concentrations having approximately stabilised since stratospheric ODSs abundances
reached their maximum at the end of the 1990s. As a result, global $O_3$ content is expected to slowly increase and return to
pre-1980 levels during the $21^{st}$ century (e.g. Weatherhead et al., 2000; Austin and Butchart, 2003; Eyring et al., 2010; WMO,
2014a, 2018). However, $O_3$ concentrations are not only affected by the presence of ODSs, but also by a wide variety of factors
(increase of greenhouse gases concentrations, changes in the Brewer-Dobson circulation and stratospheric temperatures, etc.),
making it very challenging to predict how, when and where the $O_3$ recovery will take place. In the troposphere, since $O_3$
is highly variable depending on time period, region, and proximity to fresh $O_3$ precursor emissions, there is no consistent
picture of $O_3$ tropospheric changes around the world (Steinbrecht et al., 2017; Gaudel et al., 2018; WMO, 2018). Hence, high-
quality and long-term $O_3$ measurements are essential for further improvement our understanding of $O_3$ response to natural and
anthropogenic forcings, as well as to estimate consistent trends at a global scale (Vigouroux et al., 2015; Gaudel et al., 2018;
GCOS, 2021).

Within NDACC (Network for the Detection of Atmospheric Composition Change, www.ndaccdemo.org), high-resolution
solar absorption infrared spectra have been continuously recorded since the 1990s by ground-based FTIR (Fourier Transform
InfraRed) spectrometers distributed at a global scale. By analysing the measured spectra, these instruments are capable of
providing both high-quality $O_3$ total column (TC) amounts and low-resolution $O_3$ vertical volume mixing ratio (VMR) profiles
at about twenty sites nowadays (e.g. Barret et al., 2002; Schneider and Hase, 2008; Schneider et al., 2008a; Schneider et al.,
2008b; Vigouroux et al., 2008; Viatte et al., 2011; García et al., 2012, 2014; Vigouroux et al., 2015, and references therein). In
the last years, the NDACC Infrared Working Group (IRWG, www2.acom.ucar.edu/irwg) has made considerable efforts in order
to standardise the data acquisition protocols and inversion strategies used to derive $O_3$ concentrations at the different NDACC
stations and, hence, produce uniform and consistent $O_3$ datasets (Hase et al., 2004; IRWG, 2014; Vigouroux et al., 2015).
Nonetheless, scientific discussion is still on-going seeking improvements in $O_3$ monitoring and network-wide consistency.

In this context, the present paper examines the effect of using different retrieval approaches on the quality of the FTIR
$O_3$ products, with the aim of providing an improved $O_3$ strategy that could be applied at any NDACC FTIR station. The
influence of two of the most important settings is assessed: the spectral region used for $O_3$ retrievals and simultaneous fit of the
atmospheric temperature profile. To our knowledge, so far this analysis has been approached separately in most of the studies
present in literature or has not addressed in detail yet. Previous studies have shown, for example, that an optimised selection of

the O$_3$ absorption lines or the inclusion of an additional temperature fitting significantly improve the precision of FTIR O$_3$ TCs and VMR profiles (e.g. Schneider and Hase, 2008; Schneider et al., 2008b; García et al., 2012, 2014). Nonetheless, possible combined effects were not analysed by these works.

The analysis has been performed at the O$_3$ super-site Izaña Observatory (IZO), where since 1999 ground-based FTIR observations have been carried out coincidentally to other high-quality O$_3$ measurement techniques. By using those data, a comprehensive assessment of the precision and long-term consistency of new O$_3$ retrieval strategies from ground-based FTIR spectrometry can be carried out. To this end, the current paper is structured as follows: Section 2 describes the Izaña Observatory, FTIR measurements and ancillary data considered to assess the quality of the new FTIR O$_3$ products (Brewer TC observations and Electrochemical Concentration Cell -ECC- vertical sondes). Section 3 presents the different FTIR O$_3$ retrieval strategies and their theoretical characterisation in terms of vertical sensitivity and expected uncertainty. Section 4 examines the quality and long-term reliability of the different FTIR O$_3$ TCs and VMR profiles by comparing them to the independent O$_3$ datasets. Finally, Section 5 summarises the main results and conclusions drawn from this work.

## 2 Izaña Observatory and its Ozone Programme

Izaña Observatory is a high-mountain station located on the island of Tenerife (Spain) in the subtropical North Atlantic Ocean (28.3ºN, 16.5ºW) at an altitude of 2370 m a.s.l.. The observatory is managed by the Izaña Atmospheric Research Centre (IARC, https://izana.aemet.es), which belongs to the State Meteorological Agency of Spain (AEMet, www.aemet.es). IZO is located below the descending branch of the northern subtropical Hadley cell, under a quasi-permanent subsidence regime, and typically above a stable trade wind inversion layer that acts as a natural barrier for local and regional pollution. This strategic location ensures clean air and clear-sky conditions during most of the year, making IZO an excellent station for in-situ and remote-sensing observations (Cuevas et al., 2019, and references therein).

Since many years IZO has run a comprehensive O$_3$ monitoring programme by using different measurement techniques: FTIR, Brewer, and DOAS (Differential Optical Absorption Spectroscopy) spectrometers, as well as ECC O$_3$ sondes and in-situ ultraviolet photometric analysers. The first four techniques routinely contribute to NDACC, aiming at monitoring changes in the troposphere and stratosphere with an emphasis on the long-term evolution of O$_3$ layer, while the in-situ O$_3$ records are taken in the framework of the WMO/GAW (World Meteorological Organization/Global Atmospheric Watch) programme. Refer to Cuevas et al. (2019) for more details about IZO and its atmospheric monitoring programmes.

### 2.1 FTIR Measurements

The IZO FTIR programme has been gathering high-resolution solar spectra within NDACC since 1999, when a Bruker IFS 120M spectrometer was installed by a collaborative agreement between the IMK-ASF (Institute of Meteorology and Climate Research-Atmospheric Trace Gases and Remote Sensing of Karlsruhe Institute of Technology, KIT, www.kit.edu) and AEMet-IARC. In 2005, this instrument was replaced for an upgraded model, the Bruker IFS 120/5HR, which is one of the best

performing FTIR spectrometers commercially available. For the present study, the measurements taken from 1999 to 2018, encompassing the operation of the two FTIR instruments, have been used.

Within NDACC activities, the IZO FTIR spectrometer records direct solar absorption spectra in the middle infrared spectral region, i.e., between 740 - 4250 $cm^{-1}$ (corresponding to 13.5 - 2.4 $\mu$m) by using a set of different fieldstops, narrow-bandpass filters and detectors. Nevertheless, for $O_3$ retrievals, only the 960-1015 $cm^{-1}$ spectral region is considered, which is measured with the NDACC filter 6 using a potassium bromide (KBr) beam splitter and a cooled mercury cadmium telluride (MCT) detector. The solar spectra were taken at a high spectral resolution of 0.0036 $cm^{-1}$ (250 cm maximum optical path difference, OPD, $OPD_{max}$) until April 2000, and at 0.005 $cm^{-1}$ ($OPD_{max}$=180 cm) onward. The IFS 120M's field-of-view (FOV) angle was varied between 0.17° and 0.29° depending on the measurement period, while it was always limited to 0.2° for the IFS 120/5HR. In order to increase the signal-to-noise ratio eight single scans are co-added, thereby the acquisition of one spectrum takes about 10 minutes.

NDACC FTIR solar spectra are only recorded when the line of sight (LOS) between the instrument and the sun is cloud-free. Given the IZO location, the cloud-free conditions are very common and thus FTIR measurements are typically taken about two or three times a week. For the 1999-2018 period the total number of NDACC measurement days for $O_3$ retrievals amounts to 1975, with an annual average of ~100 measurement days a year. For further details about the FTIR measurements at IZO, refer to García et al. (2021).

In order to characterise the instrumental performance of the IZO FTIR spectrometers, the ILS function has been routinely monitored about every two months since 1999 using low-pressure $N_2O$-cell measurements and LINEFIT software (v14.5), as detailed in Hase (2012). This ILS treatment ensures the independence of the FTIR trace gas retrievals and instrumental characterisation, but it also allows instrumental alignment and its temporal stability to be verified. Figure 1 depicts the time series of the ILS's modulation efficiency amplitude (MEA) and phase error (PE) parameters, between 1999 and 2018, used for $O_3$ retrievals. Three periods with different features affecting the IZO FTIR measurements can be distinguished: (1) 1999-2004, in spite of $N_2O$-cell measurements were routinely carried out, the ILS estimation is imprecise due to the instability of the IFS 120M spectrometer; (2) 2005-May 2008, the IFS 120/5HR instrument exhibits a gradual temporal drift, but the ILS function is properly assessed; and (3) June 2008-2018, the IFS 120/5HR instrument is optically aligned and well (ILS is nearly nominal). Therefore, these three periods will be independently analysed in the present work in order to examine the influence of instrument status on FTIR $O_3$ products.

## 2.2 Ancillary Data: Brewer and ECC sondes

At IZO Brewer spectrometers, managed by AEMet, have been continuously operating since 1991. In 2001, these activities were accepted by NDACC and, two years later, the RBCC-E (Regional Brewer Calibration Centre for Europe, www.rbcc-e.org) of the WMO/GAW programme was established at IZO. By recording direct solar absorption spectra in the ultraviolet spectral region, IZO RBCC-E reference instruments can provide $O_3$ TCs with a total uncertainty (standard uncertainty, k=1) between 1.2-1.5% (Gröbner et al., 2017). The high quality and long-term stability of IZO Brewer observations make them a useful reference for validating ground- and satellite-based instruments (León-Luis et al., 2018).

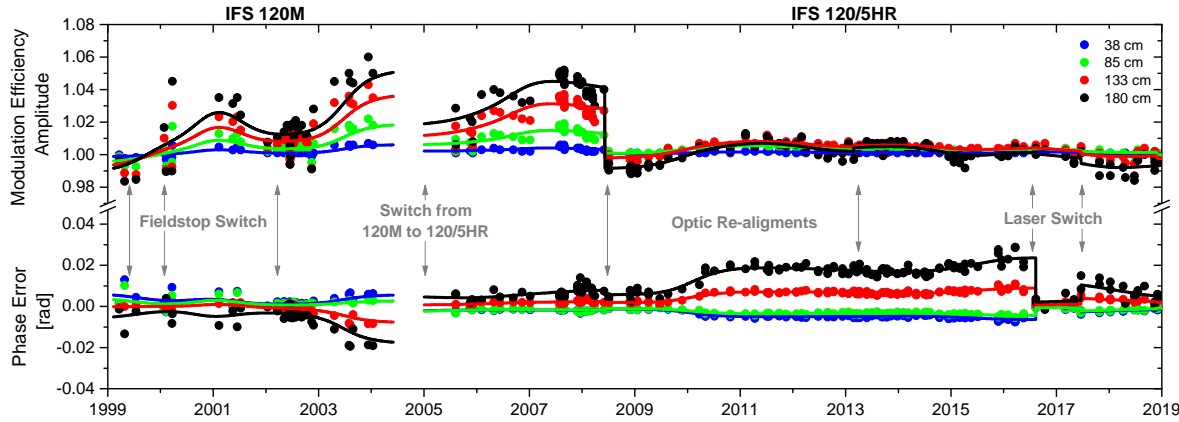

**Figure 1.** Time series of the normalised Modulation Efficiency Amplitude (MEA) and Phase Error (PE, rad) at four optical path differences (33, 85, 133, and 180 cm) for the NDACC $O_3$ measurement settings (filter 6 and MCT detector) of the IZO FTIR spectrometers between 1999 and 2018. Data points represent individual $N_2O$-cell measurements and solid lines depict smoothed MEA and PE curves. Grey solid arrows indicate punctual interventions on the instruments: different changes of fieldstops between 1999 and 2004, switch from the IFS 120M to the IFS 120/5HR system in 2005, optic re-alignments in June 2008 and February 2013, and internal laser replacements in August 2016 and June 2017.

The $O_3$ sonde programme on Tenerife started in November 1992, also run by AEMet, and since March 2001 it has operated
in the framework of NDACC. The $O_3$ sounding is based on ECC that senses $O_3$ as it reacts with a dilute solution of potassium
iodide (KI) to produce an electrical current proportional to atmospheric $O_3$ concentrations (Komhyr, 1986). The ECC sondes
(Scientific Pumps 5A and 6A) were launched once weekly from Santa Cruz de Tenerife station (30 km north-east of IZO, 36 m
a.s.l.) until 2010 and, since then, from Botanic observatory (13 km north of IZO, 114 m a.s.l.). The expected total uncertainty of
the ECC sondes is $\pm5$–$15\%$ in the troposphere and $\pm5\%$ in the stratosphere (WMO, 2014b), which is a composite of different
instrumental error contributions (i.e. sensor and background current, conversion efficiency, etc.).

Note that for the purpose of the present work both Brewer and ECC sonde databases fully cover the entire FTIR 1999-2018
period.

## 3  FTIR Ozone Observations

### 3.1  Ozone Retrieval Strategies

To analyse the influence of the spectral region and simultaneous temperature fit on the quality of FTIR $O_3$ products, six different
approaches have been defined. They combine three spectral regions and the possibility of performing or not a simultaneous
temperature retrieval (referred to as retrieval set-ups 1000, 4MWs, 5MWs, 1000T, 4MWsT and 5MWsT hereafter, Figure 2).
Set-up 1000 uses a broad spectral window covering from 1000 to 1005 cm$^{-1}$, which is the one recommended by the NDACC

IRWG (IRWG, 2014). This spectral region has been traditionally used by the FTIR community reporting high-quality $O_3$
products (e.g. Barret et al., 2002; Schneider et al., 2008a; Vigouroux et al., 2008; Lindenmaier et al., 2010; García et al., 2012;
Vigouroux et al., 2015). Set-up 5MWs uses five single micro-windows between 991 and 1014 $cm^{-1}$, which is a simplification
of the approach suggested by Schneider and Hase (2008). Schneider et al. (2008a) found that this strategy provides more precise
$O_3$ estimations than those retrieved from the broad 1000-1005 micro-window when comparing to independent measurements.
Set-up 4MWs is the same as 5MWs, but the micro-window at the greatest wavenumbers is discarded in order to avoid any
possible saturation of the strong $O_3$ absorption lines contained in this region, especially at high $O_3$ concentrations and low
solar elevations. Set-ups 1000T, 4MWsT, and 5MWsT use the same micro-windows as set-ups 1000, 4MWs, and 5MWs
respectively, but an optimal estimation of the atmospheric temperature profile is simultaneously carried out. To this end, four
$CO_2$ micro-windows are added between 962.80-969.60 $cm^{-1}$ according to García et al. (2012).

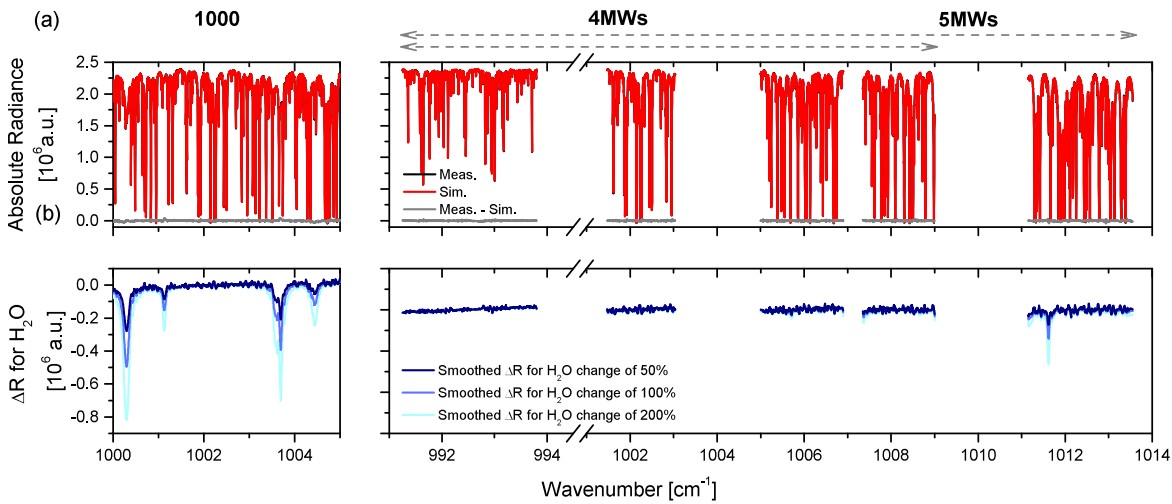

**Figure 2.** (a) Spectral regions considered in the different FTIR $O_3$ retrieval strategies: broad window used in the set-ups 1000/1000T,
encompassing the 1000-1005 $cm^{-1}$ spectral region, and the four and five micro-windows used in the set-ups 4MWs/4MWsT, between ∼991
and 1009 $cm^{-1}$, and in 5MWs/5MWsT between ∼991 and 1014 $cm^{-1}$, respectively. Red and black lines depict an example of the simulated
and measured spectrum taken on $31^{st}$ August 2007 (at solar zenith angle -SZA- of ≈50º, $O_3$ amount in the slant column -$O_3$ SC- of 390 DU,
and $H_2O$ TC of 10.7 mm). Grey lines show the difference between the measurement and simulation. (b) Spectral changes in the measured
FTIR radiances (ΔR) due to changes in $H_2O$ content of 50% (TC of 16.1 mm), 100% (TC of 21.5 mm), and 200% (TC of 32.3 mm).

With the exception of the spectral region and temperature treatment, the retrieval strategy is identical for the six approaches.
The $O_3$ VMR profiles are derived from the measured solar absorption spectra by means of PROFFIT code (PROFile FIT, Hase
et al., 2004), using an ad-hoc Tikhonov-Philips slope constraint (TP1 constraint) on a logarithmic scale. Since $O_3$ concentra-
tions are very variable around the tropopause, the logarithmic inversion has proved to be superior to the linear approach (e.g.
Schneider and Hase, 2008; Schneider et al., 2008a; Schneider et al., 2008b). Then, the $O_3$ TCs are computed by integrating

the retrieved VMR profiles from the FTIR altitude up to the top of the atmosphere. The remaining settings are based on the
NDACC IRWG recommendations (IRWG, 2014):

- The interfering species considered are $H_2O$, $CO_2$, $C_2H_4$, and the main $O_3$ isotopologues (666, 686, 668, 667 and 676 in HITRAN notation). Spectroscopic line parameters of these absorbers are taken from HITRAN 2008 (Rothman et al., 2009) with a 2009 update for $H_2O$ (www.cfa.harvard.edu).

- All set-ups use the same a-priori gas profiles, which are taken from the climatological model WACCM (Whole Atmosphere Community Climate Model, http://waccm.acd.ucar.edu)-version 6 generated by the NCAR (National Center for Atmospheric Research, J. Hannigan, personal communication, 2014).

- All set-ups apply the actual ILS time series evaluated from independent $N_2O$-cell measurements (Figure 1).

- The pressure and temperature profiles for forward simulations are taken from the NCEP (National Centers for Environmental Predictions) 12 UT daily database.

- For those approaches performing a simultaneous optimal estimation of atmospheric temperature profile, the NCEP 12 UT daily temperature profiles are used as the a-priori profiles. The a-priori temperature covariance matrix ($\mathbf{S_{aT}}$) has been constructed following Schneider et al. (2008a).

Given the importance of $H_2O$ absorption across the infrared spectrum, the treatment of $H_2O$ in $O_3$ retrievals should be carefully considered in the inversion strategy, as illustrated in Figure 2. This figure shows an example of the changes in the FTIR radiances for the spectral $O_3$ micro-windows due to changes in the $H_2O$ content of 50%, 100% and 200%. These values, which correspond to extreme conditions at IZO, could account for typical $H_2O$ content and variations at sites with greater humidity. As observed, the spectral signatures of $H_2O$ variations are much stronger in the broad 1000 spectral region than in the narrow micro-windows (4MWs/5MWs), indicating that the quality of the $O_3$ products in that region strongly depends on a correct interpretation of the spectroscopic $H_2O$ signal. Therefore, in order to minimise interference errors due to $H_2O$, a two-step inversion strategy has been applied (García et al., 2012, 2014): firstly, the actual $H_2O$ profile is derived using a dedicated $H_2O$ retrieval (Schneider et al., 2012) and, then, the $O_3$ retrieval is simultaneously performed with an $H_2O$ scaling fit, which uses the previously derived $H_2O$ state. The remaining interfering gases are simultaneously estimated with $O_3$ in the second step. As discussed in detail in Appendix A, the $H_2O$ cross-interference is reduced by the two-step strategy when the temperature retrieval is considered, which leads to this approach could be valid for humid FTIR sites.

Once FTIR retrievals are computed, they are filtered according to (1) the number of iterations at which the convergence is reached, and (2) residuals of the simulated–measured spectrum comparison. This ensures that unstable or imprecise observations are not considered (which could likely be introduced, for example, by thin clouds) (García et al., 2016). These two quality flags are applied independently on the six $O_3$ datasets, and only those measurements available for all set-ups are considered in subsequent analysis. This leads to a total of 5393 $O_3$ observations between 1999 and 2018, which are coincident and quality-filtered (∼90% of the original dataset).

## 3.2 Theoretical Quality Assessment

### 3.2.1 Vertical Sensitivity and Fitting Residuals

Because the vertical resolution of ground-based FTIR measurements is limited, a proper description of the relation between retrieved and true state must be provided together with the retrieved vertical profile. This information is theoretically characterised by the averaging kernel matrix ($\mathbf{A}$) obtained in the retrieval procedure (Rodgers, 2000). The rows of this matrix describe the altitude regions that mainly contribute to the retrieved profile and therefore the vertical distribution of the FTIR sensitivity, while its trace (also so-called "degrees of freedom for signal", DOFS) gives the number of the independent $O_3$ layers detectable by the remote-sensing FTIR instrument. As an example, Figure 3 (a) depicts the $\mathbf{A}$ rows for the 5MWs set-up for the measurement of Figure 2, while Table 1 summarises the DOFS' statistics for the six retrieval strategies considered.

**Table 1.** Summary of statistics of the DOFS and fitting residuals for the set-ups 1000/1000T, 4MWs/4MWsT, and 5MWs/5MWsT for the periods 1999-2004, 2005-May 2008, and June 2008-2018, and for the entire time series (1999-2018). Shown are median (M) and standard deviation ($\sigma$) for each period. The number of quality-filtered measurements is 519, 745, and 4219 for the three periods, respectively, and 5393 for the whole dataset. The strategies showing the best performance, in terms of largest DOFS and smallest residuals, are highlighted in bold for each period.

| | DOFS | | | | Residuals ($\times 10^{-3}$) | | | |
| --- | --- | --- | --- | --- | --- | --- | --- | --- |
| | 1999-2004 | 2005-2008 | 2008-2018 | 1999-2018 | 1999-2004 | 2005-2008 | 2008-2018 | 1999-2018 |
| Set-up | M, $\sigma$ | M, $\sigma$ | M, $\sigma$ | M, $\sigma$ | M, $\sigma$ | M, $\sigma$ | M, $\sigma$ | M, $\sigma$ |
| 1000 | 3.76, 0.25 | 3.99, 0.21 | 3.98, 0.14 | 3.97, 0.18 | 3.56, 1.90 | 2.62, 0.93 | 2.75, 0.55 | 2.77, 0.93 |
| 4MWs | 4.09, 0.28 | 4.34, 0.12 | 4.30, 0.12 | 4.30, 0.17 | 3.51, 1.90 | 2.57, 0.85 | 2.70, 0.54 | 2.71, 0.91 |
| 5MWs | **4.29**, 0.28 | **4.56**, 0.15 | **4.52**, 0.12 | **4.51**, 0.17 | 3.53, 1.90 | 2.58, 0.87 | 2.70, 0.55 | 2.73, 0.92 |
| 1000T | 3.66, 0.31 | 3.98, 0.26 | 3.92, 0.17 | 3.91, 0.22 | 3.50, 1.90 | 2.60, 0.91 | 2.73, 0.55 | 2.75, 0.92 |
| 4MWsT | 3.91, 0.33 | 4.24, 0.18 | 4.17, 0.15 | 4.16, 0.20 | 3.46, 1.90 | **2.55**, 0.86 | **2.68**, 0.54 | **2.70**, 0.91 |
| 5MWsT | 4.10, 0.33 | 4.42, 0.20 | 4.35, 0.15 | 4.35, 0.21 | **3.45**, 1.90 | 2.56, 0.87 | **2.68**, 0.54 | **2.70**, 0.91 |

The $\mathbf{A}$ rows are quite similar for all set-ups with a median total DOFS value of $\sim4$, meaning that the FTIR system is able to roughly resolve four independent atmospheric $O_3$ layers: the troposphere (2.37–13 km), the upper troposphere/lower stratosphere (UTLS) or tropopause region (12–23 km), the middle stratosphere around the ozone maximum (22–29 km), and the upper stratosphere (28–42 km). However, the total DOFS values are found to be greater for those set-ups using narrower micro-windows than for the broad spectral window (see Table 1), whereby the former configurations seem to offer a better vertical sensitivity (especially the 5MWs set-up). This pattern is independent on the FTIR instrument and consistent over time, as observed for the three periods analysed (1999-2004, 2005-May 2008, and June 2008-2018). The comparison between the instruments also reveals, as expected, the lower sensitivity to the $O_3$ concentrations of the IFS 120M spectra as compared to the IFS 120/5HR measurements. The total DOFS values overall differ by 5.5% (1000) and 5.1% (5MWs) between the

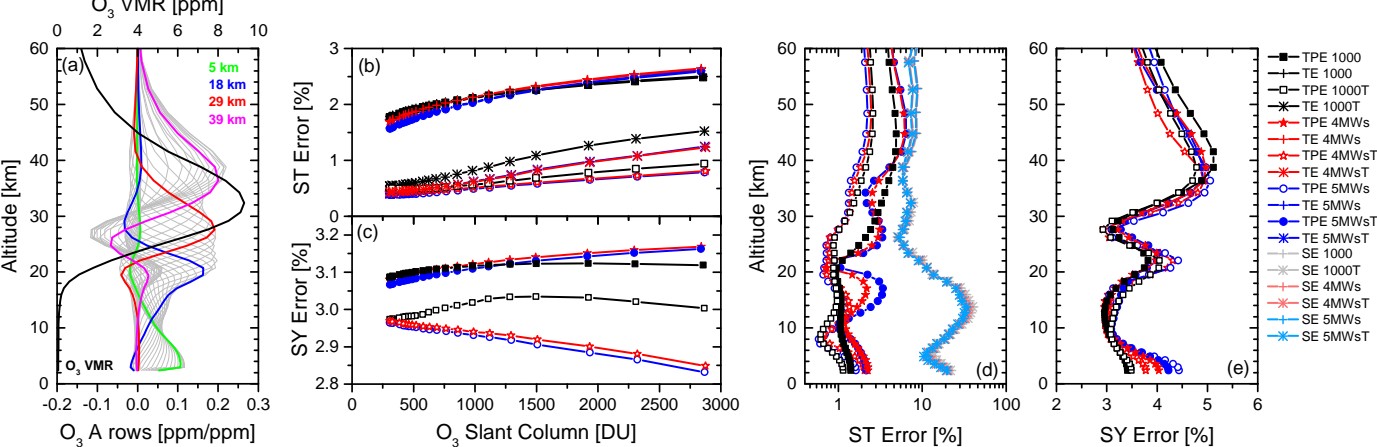

**Figure 3.** Summary of the theoretical quality assessment. (a) Example of averaging kernel (**A**) rows for the 5MWs set-up on a logarithm scale for the measured spectrum of Figure 2. Coloured lines highlight the **A** rows at the altitudes of 5, 18, 29 and 39 km, which are representative for the four layers detectable by the FTIR instrument (total DOFS of 4.41). The retrieved $O_3$ VMR profile is also shown. (b) Statistical (ST) and (c) systematic (SY) contribution of the total parameter errors (TPE, in %) and total error (TE, in %) for $O_3$ TCs retrieved from the set-ups 1000/1000T, 4MWs/4MWsT, and 5MWs/5MWsT as a function of $O_3$ slant column [DU] for measurements taken on $31^{st}$ August 2007 from SZAs between 84º ($\sim$07:00 UT) and 21º ($\sim$13:30 UT). TPE is computed as the square root of the quadratic sum of all ST and SY error sources considered with the exception of the smoothing error (SE), while TE considers TPE and SE. Example of ST (d) and SY (e) contribution of the TPE and SE profiles [%] for all set-ups for the measured spectrum of Figure 2.

1999-2004 and 2008-2018 periods, respectively. When simultaneously fitting the atmospheric temperature profile, the median
DOFS values slightly decrease for all strategies, because the information contained in measured spectra is then split into $O_3$ and temperature retrievals (the retrieved state vector space is not perfectly orthogonal). Likewise, the differences between both instruments become more accentuated (e.g. 6.6% and 5.7% between the 1999-2004 and 2008-2018 periods for 1000T and 5MWsT, respectively). As with DOFS analysis, the fitting residuals are smaller for those set-ups that use narrow micro-windows and apply the temperature fit, allowing a more detailed interpretation of measured spectra to be obtained (as also
summarised in Table 1). In addition, the IFS 120M retrievals are found to be considerably more variable than IFS 120/5HR data. However, it is fair to admit that the differences among retrieval strategies lie within the respective error confidence intervals, therefore no robust conclusions can be reached. Note that, in order to make a fair comparison, the fitting residuals are computed as the noise-to-signal ratio for a common spectral region contained in all set-ups (1001.47-1003.04 cm$^{-1}$).

### 3.2.2 Uncertainty Analysis

The characterisation of the different FTIR $O_3$ products has been completed by an uncertainty analysis, which evaluates how different error sources could be propagated into the retrieved products. The theoretical error assessment carried out in the

present paper is based on Rodgers (2000), and analytically performed by PROFFIT package. The Rodgers' formalism distinguishes three types of error: (1) smoothing error (SE) associated with the limited vertical sensitivity of the remote-sensing FTIR instruments, (2) spectral measurement noise, and (3) uncertainties in the input/model parameters (instrumental characteristics, spectroscopy data, ...), which are split into statistical (ST) and systematic (SY) contributions. Given that SE can be considered as an inherent characteristic of the remote-sensing technique, it is not included in the uncertainty assessment suggested by the NDACC IRWG (IRWG, 2014). Therefore, it has been separately considered in this work by distinguishing the total parameter error (TPE), which is calculated as the the square root of the quadratic sum of all error sources considered with the exception of SE, and the total error (TE), which considers both TPE and SE. A detailed description of the uncertainty assessment is given in Appendix B.

In order to assess the effect of $O_3$ absorption signatures on the uncertainty budget, the dependence of the estimated errors on $O_3$ slant column (SC) amounts for each retrieval set-up has been examined. This analysis would allow possible inconsistencies between the set-ups or saturations of $O_3$ absorption lines at high $O_3$ concentrations to be detected. As shown in Figure 3, both statistical and systematic uncertainties do depend on $O_3$ spectroscopic signatures for all set-ups due to the increase of most of the error sources considered at larger $O_3$ SCs (see details in Appendix B). Figure 3 also documents that the inclusion of a simultaneous temperature retrieval significantly improves the theoretical performance for all FTIR $O_3$ products. Although this fit generates a negative cross-interference with the ILS, measurement noise and smoothing error (especially for the 1000T set-up, Figure B1), the temperature error contribution is in return nearly eliminated, leading to the total ST budget decreasing by ∼1%. TPE and TE range from 1.5 to 2.5% (between 250 and 3000 DU) for set-ups without a simultaneous temperature fit, while TPE varies from 0.5 to 1.0% (to 1.5% for TE due to the influence of SE) when including the temperature retrieval. The total systematic contributions also drop by 0.3% (at high $O_3$ SCs) with mean values of ∼3%. It is worth highlighting that an inconsistency between 4MWsT/5MWsT and 1000T set-ups has been detected in the systematic uncertainty budget, which is determined by the spectroscopy errors. For the 1000T configuration, the spectroscopic SY error exhibits a reverse smile curve with $O_3$ concentrations, and is considerably greater than for the narrow micro-window set-ups. This result might point to a possible saturation of the deeper $O_3$ lines contained in the broad window or some inconsistency in the spectroscopy parameters. For example, an erroneous parameterisation of the temperature dependence of the $O_3$ line width may produce systematic differences between actual and retrieved temperature profiles (Schneider and Hase, 2008), therefore affecting the absolute value of $O_3$ FTIR products.

Vertically, the most important contribution is SE reaching ∼40% in the UTLS region (Figure 3 (d)), where the $O_3$ concentrations are very variable and the profile might be highly structured. The FTIR system is not able to resolve such fine vertical structures. Excluding SE, the statistical TPE profiles are strongly linked to the atmospheric temperature with maximal errors beyond the UTLS region (where the maximum FTIR sensitivity and the largest $O_3$ concentrations are also located, see Figure 3 (a)). This pattern is consistently observed for both set-ups with and without fitting the atmospheric temperature profiles. However, the error values drastically drop when considering the temperature in the retrieval procedure. TPE values between 1.0-1.5% are expected for the 1000T, 4MWsT and 5MWsT set-ups, and as high as 6% when the temperature fit is not taken into account. In relation to the systematic uncertainty profiles (Figure 3 (e)), they range from 3% in the UTLS and middle strato-

sphere (around 30 km) to 5% in the upper stratosphere. As with statistical errors, temperature contribution decreases when including the atmospheric temperature profile in the retrieval, leading to smaller systematic errors in $O_3$ TCs as mentioned above.

The error estimation presented here assumes the same set of uncertainty values for all set-ups, which is representative of the IFS 120/HR instrument in the period 2005-2008 (Table B1 in Appendix B). However, some error sources do strongly depend on instrument status (particularly the ILS function, solar pointing, and measurement noise), affecting the total uncertainty budget. In order to account for the different quality periods of the IZO FTIR instruments, the uncertainty analysis for different set of error values has been included in Appendix B.

To summarise, using several narrow micro-windows instead of a single broad region and applying a temperature profile fit has been found to provide more precise FTIR $O_3$ estimations by increasing the vertical sensitivity and decreasing the expected uncertainties. The simultaneous temperature retrieval could be a suitable approach provided the FTIR system is properly characterised, with a continuous assessment of ILS function, and stable over time (e.g. IFS 120/5HR spectrometers), in order to minimise the negative influence of the ILS uncertainties and measurement noise on $O_3$ retrievals. Finally, although the

narrow micro-windows set-ups provide very consistent results, the 5MWsT set-up has theoretically shown to be superior for the typical $O_3$ concentrations observed at the tropical and subtropical latitudes.

## 4    Comparison to Reference Observations

### 4.1    FTIR and Brewer Ozone Total Columns

The performance of the six FTIR $O_3$ retrieval strategies has been assessed by comparison with coincident NDACC Brewer $O_3$

TC data. In order to mitigate the influence of the $O_3$ intra-day variations on comparisons, only FTIR and Brewer measurements within a temporal coincidence of 5 minutes have been paired, which makes a total of 2231 coincidences between 1999 and 2018.

Figure 4 displays the time series of the Brewer observations together with four examples of FTIR retrievals (for simplicity only the set-ups 1000/1000T and 5MWs/5MWsT are depicted), as well as the time series of corresponding relative differences

(RD, FTIR-Brewer). The temporal $O_3$ TCs variations are in general reproduced well by all FTIR products. However, the different performance of the two FTIR instruments becomes evident from this figure: while the RD values of the IFS 120/5HR instrument are very stable over time, the IFS 120M instrument exhibits a more erratic behavior. Besides the greater variability of the IFS 120M and the switch of instrument in 2005, the most remarkable feature is a discontinuity detected at the beginning of 2010 by a non-parametric change-point test (Lanzante, 1996) (at 99% confidence level). The systematic jump is of ∼1.1%

for the set-ups without fitting temperature, and is partially corrected by retrieving temperature, ∼0.7%. The change-point, already reported by García et al. (2014), is likely due to modifications on the IFS 120/5HR spectrometer (failure of the interferometer's scanner motor and its subsequent replacement). Another change-point was detected by the Lanzante's approach around the beginning of 2014, but of less intensity. It is worth highlighting that both discontinuities were also detected in the differences of $O_3$ TCs retrieved from the different FTIR set-ups, especially by those including the temperature fit. Hence, when

no independent observations are available, the analysis of different FTIR products could offer additional tools for identifying inconsistencies and documenting the long-term instrumental stability.

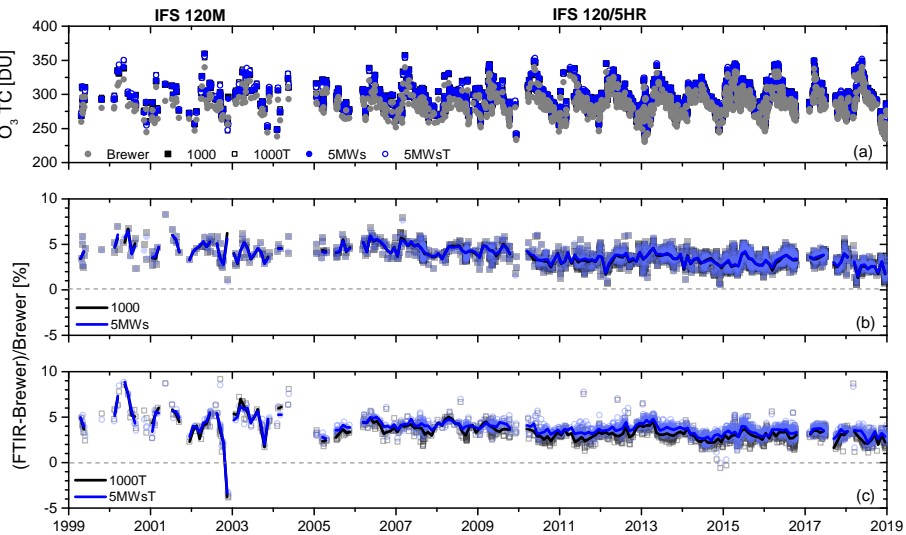

**Figure 4.** Summary of FTIR-Brewer comparison from 1999 to 2018. (a) Time series of $O_3$ TCs [DU] as observed by Brewer and FTIR 1000/1000T, and 5MWs/5MWsT set-ups. (b) Time series of the relative differences RD for the set-ups 1000 and 5MWs, which are calculated as RD[%]=$100 \times$($O_3$ $TC_X$ - $O_3$ $TC_Y$)/$O_3$ $TC_Y$, where X and Y means FTIR and Brewer, respectively. (c) As for (b), but for the set-ups 1000T and 5MWsT. Solid lines in (b) and (c) correspond to monthly medians.

Figure 4 also reveals that, although the scatter of RD is significantly improved by the temperature retrieval, these strategies present in general more extreme values as compared to the Brewer data. The RD range between the percentiles $0.1^{st}$ and $99.9^{th}$ is 7.4% and 7.0% for the 1000 and 5MWs set-ups, respectively, while it is 9.3% and 8.6% for the 1000T and 5MWsT
configurations, respectively. Note that, for example, the extreme RD values obtained for 1000T and 5MWsT set-ups at the beginning of 2000 and at ending of 2002 (Figure 4 (c)) are not reproduced by the 1000 and 5MWs strategies (Figure 4 (b)). This pattern is consistently observed for all set-ups and over time. The extreme RD values may indicate measurement days with an unusual temperature vertical stratification, which might be wrongly captured by the Brewer and FTIR products assuming a fixed temperature (and pressure) profile. For forward calculations of those FTIR strategies without a simultaneous
temperature fit, the temperature and pressure profiles are updated daily from the NCEP database, as previously mentioned, but they are kept constant during the $O_3$ retrieval procedure. Regarding Brewer, no temperature or pressure dependence is considered in the operational data processing (Redondas et al., 2014; Rimmer et al., 2018). Particularly, the Brewer $O_3$ TCs are computed using the so-called effective $O_3$ cross sections throughout the atmosphere (Bass and Paur, 1985), corresponding to an $O_3$ effective height of 22 km and a fixed effective temperature of the $O_3$ layer of -45ºC. These simplifications can produce
systematic (seasonal dependence) and random errors (Redondas et al., 2014; Gröbner et al., 2021). In fact, at IZO the effective

temperature and $O_3$ height significantly differ from the assumed values by Brewer processing in winter months, when the extreme RD values are overall observed (Figure 4 (c)). Nevertheless, a more dedicated study would be desirable to deeply investigate the causes driving these anomalous values.

When analysing in detail the intercomparison results (Table 2), it is confirmed that, first, the more refined set-ups using narrow micro-windows offer the best performance (especially 5MWs/5MWsT) independently of the treatment of the atmospheric temperature profile, and second, the effect of the simultaneous temperature fit on the FTIR $O_3$ quality depends on the instrumental stability. The agreement between the FTIR and Brewer observations significantly worsens for the more unstable IFS 120M spectrometer when the temperature fit is included in the retrieval procedure (largest median bias and scatter, and least correlation for the 1999-2004 period). Opposite behaviour is documented for the IFS 120/5HR periods: the temperature retrieval consistently improves the precision and accuracy of all FTIR $O_3$ products by considerably reducing the dispersion and bias of the RD distributions. Thus, the best performance is obtained by those set-ups using narrow micro-windows, with a RD scatter of only ∼0.6-0.7% (it increases up to ∼0.8% for the broader region set-up) for the IFS 120/5HR instrument; while it is as high as 2% for the IFS 120M when the simultaneous temperature fit is carried out. These values perfectly agree with previous studies (e.g. Schneider et al., 2008a; García et al., 2012), and lie within the expected precision of both instruments (see Sections 2.2, and 3.2.2, and Appendix B). In fact, as shown in Appendix B, total statistical errors of ∼0.5-1.5% can be expected depending on the instrument status (i.e. the instrumental degradation and solar pointing errors). Nonetheless, Table 2 also documents that the scatter found in RD is noticeably lower than that predicted when the temperature fit is not considered, especially for the IFS 120/5HR instrument (see $M_{TE}$ values). This fact could point out that sources of uncertainty partially cancel each other out, which is not fully reproduced by the theoretical error assessment, and/or the possible overestimation of the assumed temperature uncertainty. As also shown in Appendix B, reducing the latter would contribute to reconciling the experimental and theoretical results. Note that the scatter values of the IFS 120/5HR spectrometer could be a conservative value for the precision of FTIR $O_3$ TC estimations, since they can be interpreted as the root-squares-sum of the precision of Brewer and FTIR instruments.

Regarding the systematic differences, a median bias of ∼3-5% is obtained. Such discrepancies are consistent with previous studies (e.g. Schneider et al., 2008a; García et al., 2012; García et al., 2016) and mainly attributed to inconsistencies between infrared and ultraviolet spectroscopic parameters (e.g. Piquet-Varrault et al., 2005; Gratien et al., 2010; Drouin and Yu, 2017; Tyuterev et al., 2019). In fact, as recently presented by Gordon et al. (2022), the most recent release of HITRAN spectroscopic database (HITRAN 2020) improves the $O_3$ line intensities in the 1000 $cm^{-1}$ spectral band by applying a scaling factor of 3%. This correction agrees well with our theoretical uncertainty estimations and the overestimation found for the FTIR products.

Similar conclusions can be reached in general when comparison is performed as a function of $O_3$ signatures in the slant path (Figure 5). The temperature fit improves the performance of the stable instrument and makes it worse for the more unstable instrument, independently of $O_3$ SC range covered at IZO. In addition, the bias between FTIR and Brewer data overall decreases by ∼1% as $O_3$ SCs increase for all set-ups (Figure 5 (a), (c) and (e)). This dependence can in part be accounted for the Brewer systematic uncertainties in the absolute calibration process, which are amplified for low $O_3$ SCs (up to 0.5%, Schneider et al. (2008a)). However, for the more stable IFS 120/5HR period (Figure 5 (e)), the bias alters this

**Table 2.** Summary of statistics for FTIR-Brewer comparison for the set-ups 1000/1000T, 4MWs/4MWsT, and 5MWs/5MWsT: median (M, in %) and standard deviation ($\sigma$, in %) of the relative differences, and Pearson correlation coefficient (R) of the direct comparison for the periods 1999-2004, 2005-May 2008 and June 2008-2018, and for the entire time series (1999-2018). Shown are also the median of theoretical TPE ($M_{TPE}$), SE ($M_{SE}$), and TE ($M_{TE}$) uncertainties. The number of coincident FTIR-Brewer measurements is 93, 185, and 1892 for the three periods, respectively, and 2170 for the whole dataset. The strategies showing the best performance, in terms of smallest $\sigma$ and $M_{TE}$, are highlighted in bold for each period.

| Set-up | 1999-2004<br>M, $\sigma$, R, $M_{TPE}$, $M_{SE}$, $M_{TE}$<br>[%], [%], - , [%], [%], [%] | 2005-2008<br>M, $\sigma$, R, $M_{TPE}$, $M_{SE}$, $M_{TE}$<br>[%], [%], - , [%], [%], [%] | 2008-2018<br>M, $\sigma$, R, $M_{TPE}$, $M_{SE}$, $M_{TE}$<br>[%], [%], - , [%], [%], [%] | 1999-2018<br>M, $\sigma$, R, $M_{TPE}$, $M_{SE}$, $M_{TE}$<br>[%], [%], - , [%], [%], [%] |
|---|---|---|---|---|
| 1000 | 4.29, 1.38, 0.957, 1.85, 0.23, 1.87 | 4.47, 0.86, 0.970, 1.85, 0.23, 1.87 | 3.35, 0.83, 0.982, 1.83, 0.23, 1.85 | 3.46, 0.95, 0.975, 1.84, 0.23, 1.85 |
| 4MWs | 4.28, 1.36, 0.959, 1.79, 0.19, 1.80 | 4.49, 0.85, 0.971, 1.78, 0.19, 1.80 | 3.34, 0.82, 0.982, 1.77, 0.19, 1.78 | 3.45, 0.93, 0.976, 1.77, 0.19, 1.78 |
| 5MWs | 4.35, **1.32**, 0.962, 1.68, 0.16, 1.68 | 4.53, 0.82, 0.973, 1.66, 0.17, 1.67 | 3.41, 0.81, 0.983, 1.65, 0.17, 1.66 | 3.50, 0.91, 0.977, 1.65, 0.17, 1.66 |
| 1000T | 4.83, 1.97, 0.926, 0.52, 0.24, 0.57 | 3.79, 0.82, 0.972, 0.51, 0.15, 0.53 | 3.04, 0.73, 0.986, 0.51, 0.16, 0.53 | 3.12, 0.90, 0.977, 0.51, 0.16, 0.53 |
| 4MWsT | 4.84, 1.90, 0.934, 0.44, 0.21, 0.48 | 3.97, 0.66, 0.981, 0.42, 0.10, 0.44 | 3.32, 0.68, 0.988, 0.42, 0.12, 0.44 | 3.40, 0.83, 0.981, 0.42, 0.12, 0.44 |
| 5MWsT | 4.81, 1.82, 0.940, 0.40, 0.22, **0.44** | 4.15, **0.63**, 0.983, 0.39, 0.09, **0.40** | 3.44, **0.67**, 0.988, 0.39, 0.11, **0.40** | 3.53, **0.81**, 0.982, 0.39, 0.11, **0.40** |

behaviour for $O_3$ SCs beyond ~550 DU for set-ups using narrow micro-windows when the simultaneous temperature retrieval is considered. This issue could be attributed to inconsistencies in the spectroscopic parameters at higher wavenumbers, which gain importance as $O_3$ concentrations increase, and is in line with the theoretical systematic inconsistency found between the 1000T and 4MWsT/5MWsT set-ups (Section 3.2.2). However, the number of FTIR-Brewer coincidences at IZO is rather small

for $O_3$ SCs greater than 550 DU (i.e. less than ~20% of Brewer data in the 1999-2018 period), therefore a more robust dataset would be recommendable to better understand what drives this different pattern.

     As pointed out by the theoretical uncertainty analysis, statistical errors are expected to increase with $O_3$ SCs for all set-ups. This can be seen in the scatter of RD for the IFS 120/5HR periods (Figure 5 (d) and (f)) when the temperature fit is not applied (i.e. the scatter of RD increases by 0.2% in the 2008-2018 period at larger $O_3$ SCs). However, the intercomparison results

seem not to exhibit a similar dependence when the temperature retrieval is considered, as expected. Including or not this fit can result in differences of 0.2% at larger $O_3$ SCs. This could be attributed to an underestimation of the ILS and/or baseline errors in the theoretical assessment: the ILS contribution to total uncertainty budget decreases as $O_3$ SCs increase and becomes more important when the temperature fit is applied (see Figure B1). The same behaviour is found for baseline error (data not shown). Hence, an increment in the assumed uncertainties for these two error sources, when the temperature takes part of

the retrieval procedure, could reduce partially discrepancies between the theoretical and experimental assessment. Note that while 1000/1000T set-ups provide the most accurate $O_3$ TCs with respect to Brewer data for the whole $O_3$ SC range for both instruments, the 5MWs/5MWsT offer the most precise $O_3$ TCs. This result further corroborates that the broad region seems to be less sensitive to the improvement generated by the temperature retrieval.

     The long-term FTIR time series used in this study allows us to investigate the overall quality and long-term consistency

of new products, but also the effects at different timescales. At a seasonal scale, the agreement between FTIR and Brewer is

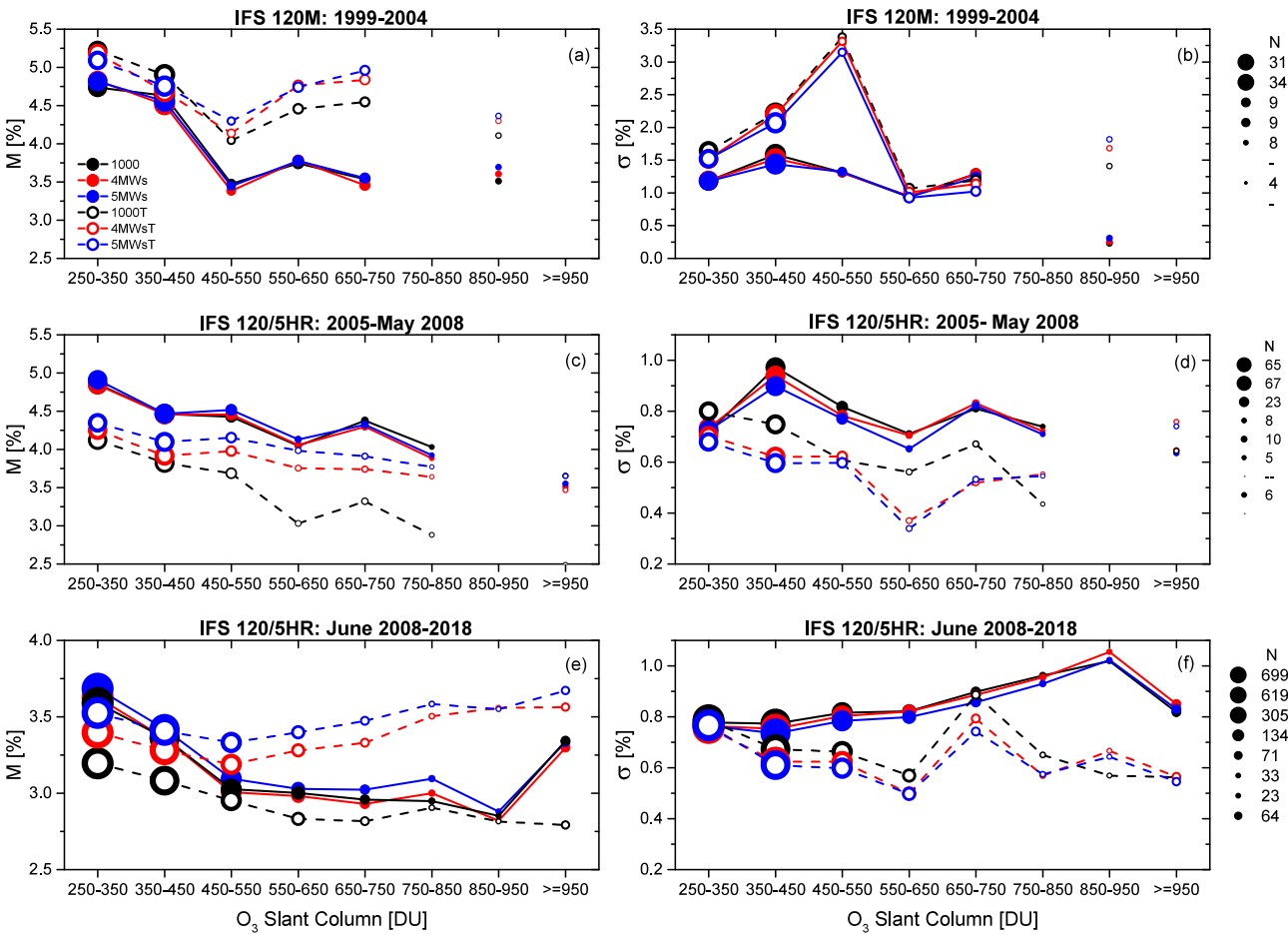

**Figure 5.** Median and standard deviation of relative differences RD (FTIR-Brewer) with respect to Brewer $O_3$ slant column [DU] for the periods 1999-2004, 2005-May 2008, and June 2008-2018. (a), (c), and (e) shows median RD (M, in %) values for the three periods, respectively, and (b), (d), and (f) the same, but for standard deviation of RD ($\sigma$, in %). Dotted area indicates the number of coincident FTIR-Brewer measurements for each $O_3$ SC interval (N), which are included in the legend of each subplot. For a better visualisation a scale factor of 3, 2, and 1 was applied to dotted area for the periods 1999-2004, 2005-May 2008, and June 2008-2018, respectively.

excellent: the annual cycles are completely in phase with Pearson correlation coefficients greater than 0.99 for all retrieval strategies considered (Figure 6 (a) and (b)). However, the bias between both techniques depends on $O_3$ amounts leading to a seasonal effect on RD, which is likely due to the fact that the Brewer and FTIR products exhibit different response to $O_3$ seasonal variations. On the one hand, the FTIR sensitivity is strongly anti-correlated with the $O_3$ SC annual cycle: the less the $O_3$ amounts, the less saturated the $O_3$ absorption lines. This results in minimum (maximum) DOFS in winter (spring/summer) (García et al., 2012). On the other hand, the different treatment of the atmospheric temperature and $O_3$ vertical distribution

in the Brewer and FTIR data processing also generates seasonal artefacts, as stated above for the extreme RD values. In fact, including the temperature retrieval significantly modifies the RD seasonal patterns, as observed in Figure 6 (c) and (d).

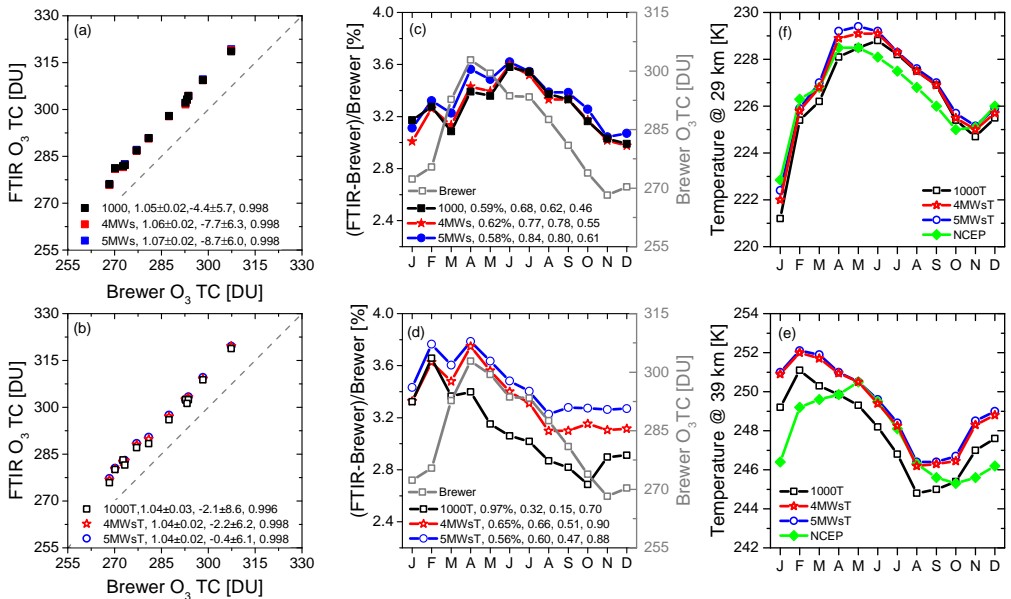

**Figure 6.** Summary of the FTIR-Brewer comparison at a seasonal scale for the 2009-2018 period. (a) and (b) scatter plots of the Brewer and FTIR O$_3$ TCs for the set-ups 1000/4MWs/5MWs and 1000T/4MWsT/5MWsT, respectively. (c) and (d) averaged annual cycle of Brewer O$_3$ TCs and RD for the set-ups 1000/4MWs/5MWs, and 1000T/4MWsT/5MWsT, respectively. (e) and (f) averaged annual cycle of the atmospheric temperature at 29 km and 39 km, respectively, retrieved from the set-ups 1000T, 4MWsT, and 5MWsT, as well as NCEP database. Legend in (a) and (b) displays the slope, offset and Pearson correlation coefficient of the least-square fits, and in (c) and (d) shows the amplitude of RD annual cycle (in %), and Pearson correlation coefficients between RD and Brewer O$_3$ TC annual cycles, and atmospheric temperature (at 29 km and 39 km) annual cycles, respectively.

For those approaches without a simultaneous temperature fit (Figure 6 (c)), the RD annual cycle seems to follow the typical
O$_3$ TC seasonality at the subtropical latitudes: peak values in spring and minimum in autumn–winter, as a result of the joint effect of the annual shift in the height of subtropical tropopause, and annual cycle of the O$_3$ photochemical production associated to tropical insolation (e.g. García et al., 2014; García et al., 2021). Hence, a significant correlation between the averaged RD and O$_3$ TC annual cycles for all set-ups is found, with Pearson correlation coefficients ranging from 0.68 to 0.84 for the 1000 and 5MWs strategies, respectively. This relationship drops to correlation values between 0.32 and 0.60 for the 1000T and
5MWsT set-ups, respectively. However, in return, a seasonal dependence on the upper–stratospheric temperature is detected (Figure 6 (f) displays, as an example, the averaged temperature annual cycle at 39 km from the NCEP database along with those retrieved from the FTIR set-ups). The correlation between the averaged annual cycles of the upper-stratospheric temperature and RD is ∼0.70 and ∼0.90 for the broad and narrow micro-window set-ups, respectively, including the temperature fit; while

it is limited between 0.46 and 0.61 when the temperature retrieval is not considered. Note that a subtle relationship is found with the temperature in the middle or lower stratosphere (e.g. at 29 km in Figure 6 (e)). Additionally, it has been found that the RD seasonal amplitudes are overall augmented by the temperature retrieval. The broad spectral region seems to be the most sensitive to this effect: the RD peak-to-peak amplitude goes from 0.59% (1000) to 0.97% (1000T), while it is modified less than 0.05% for the 4MWs/5MWs set-ups.

## 4.2 FTIR and ECC Ozone Vertical Profiles

In order to evaluate the influence of the six retrieval strategies on $O_3$ vertical distribution, Figure 7 displays the vertical profiles of the relative differences between FTIR and ECC sondes for the three periods considered, while Table 3 summarises the comparison for the $O_3$ layers that are sufficiently detectable well by the FTIR system, i.e., the partial column (PC) between 2.37–13 km, 12–23 km and 22–29 km (the DOFS for all these layers is typically larger than one). For this comparison, the approach suggested by Schneider et al. (2008b) and García et al. (2012) was followed, whereby the ECC sondes were corrected daily by comparing them to coincident Brewer data. By means of this correction, the quality and long-term stability of the ECC sonde data can be significantly improved. In addition, the highly-resolved ECC profiles ($x_{ECC}$) were vertically-degraded ($\hat{x}_{ECC}$) applying the averaging kernels obtained in the FTIR $O_3$ retrieval procedure (Rodgers, 2000), as follows:

$$\hat{x}_{ECC} = \mathbf{A}(x_{ECC} - x_a) + x_a, \tag{1}$$

where $x_a$ is the a-priori $O_3$ VMR profiles. The ECC smoothing allows, on the one hand, the limited sensitivity of FTIR data to be taken into account and, on the other hand, the effect of the different strategies on retrieved $O_3$ profiles to be directly assessed. Note that, in order to homogenise the comparison, only the ECC sondes with continuous measurements up to 29 km have been considered. Beyond this altitude, the ECC data were completed using the a-priori profiles used in FTIR $O_3$ retrievals for computing $\hat{x}_{ECC}$. Finally, the temporal collocation window between FTIR and ECC sondes is extended to $\pm 3$ hours around the sonde launch (typically at 12 UT) to ensure sufficient pairs for a robust comparison (N=272 in the 1999-2018 period).

The RD profiles show a strong vertical stratification, whereby the three independent layers detectable by the FTIR systems up to ∼30 km are clearly discernible (troposphere, UTLS, and middle stratosphere, recall Figure 3). Particularly, beyond the UTLS region, the influence of ILS uncertainties on the retrieved $O_3$ profiles is getting importance with altitude, since the full width at half maximum of the narrow $O_3$ absorption lines and ILS function becomes to be comparable. For all set-ups and periods, the simultaneous temperature fit has turned out to worsen the agreement between FTIR and ECC sondes at higher altitudes (see standard deviation profiles in Figure 7 (b), (e), and (h)). However, as the instrument is better aligned and more stable over time, the effect of this cross-interference becomes less significant until no noticeable differences are observed for the 2008-2018 period. For example, the scatter at 29 km for FTIR-ECC comparison only changes from 3.6% to 3.8% for the 5MWs and 5MWsT set-ups, respectively, for the 2008-2018 period, while the variation ranges between 4.3% and 7.5% for the 1999-2004 period. In the UTLS region the best agreement is overall found for those set-ups without fitting temperature profile. This may indicate the prevalence of negative effect of the ILS uncertainties and measurement noise over the improvement

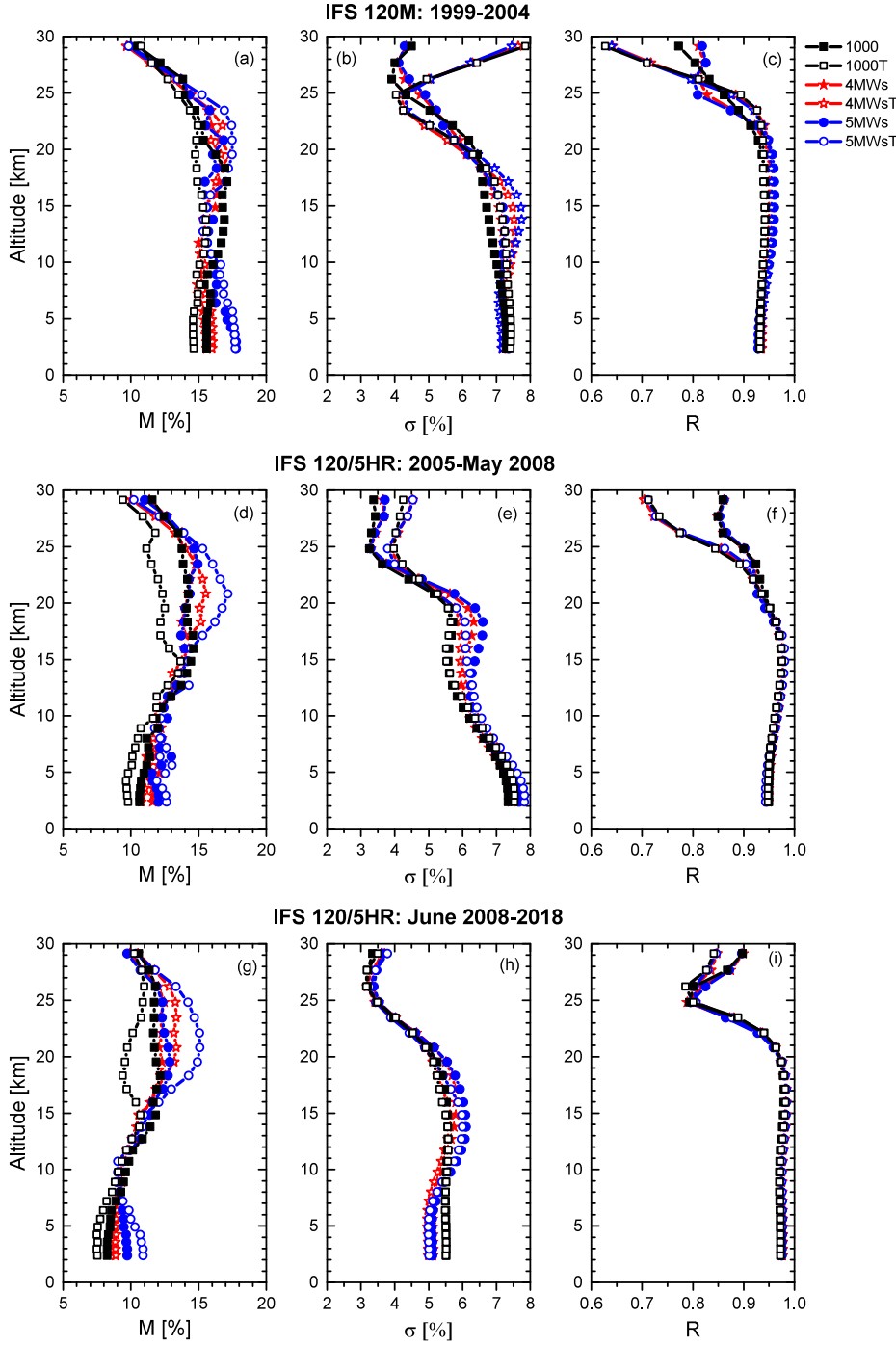

**Figure 7.** Summary of FTIR-smoothed ECC comparison for the periods 1999-2004, 2005-May 2008, and June 2008-2018. (a), (d), and (g) display the vertical profiles of median (M) RD (FTIR-ECC, in %) for the three periods, respectively. (b), (e), and (h) as for (a), (d), and (g), but for standard deviation of RD distributions ($\sigma$, in %). (c), (f), and (i) as for (a), (d), and (g), but for Pearson correlation coefficient. The number of coincident FTIR-ECC measurements is 56, 49, and 167 for the periods 1999-2004, 2005-May 2008, and June 2008-2018, respectively.

**Table 3.** Same as Table 2, but for FTIR-smoothed ECC comparison for the $O_3$ partial columns computed between 2.37-13 km, 12-23 km, and 22-29 km. The number of coincident FTIR-ECC measurements is 56, 49, and 167 for the three periods, respectively, and 272 for the whole dataset. The strategies showing the best performance, in terms of smallest $\sigma$ and $M_{TE}$, are highlighted in bold for each period.

| Set-up | 1999-2004 $M, \sigma, R, M_{TPE}, M_{SE}, M_{TE}$ [%], [%], - , [%], [%], [%] | 2005-2008 $M, \sigma, R, M_{TPE}, M_{SE}, M_{TE}$ [%], [%], - , [%], [%], [%] | 2008-2018 $M, \sigma, R, M_{TPE}, M_{SE}, M_{TE}$ [%], [%], - , [%], [%], [%] | 1999-2018 $M, \sigma, R, M_{TPE}, M_{SE}, M_{TE}$ [%], [%], - , [%], [%], [%] |
|---|---|---|---|---|
| **FTIR-ECC at 2.37-13 km** | | | | |
| 1000 | 15.86, 7.08, 0.934, 1.14, 6.34, 6.44 | 11.44, 6.58, 0.956, 1.09, 5.93, 6.03 | 9.08, 5.46, 0.970, 1.06, 5.68, 5.78 | 10.63, 6.61, 0.953, 1.07, 5.76, 5.86 |
| 4MWs | 15.08, 7.11, 0.937, 1.26, 5.76, 5.91 | 12.08, **6.54**, 0.958, 1.28, 6.32, 5.48 | 9.38, 5.10, 0.974, 1.25, 5.17, 5.33 | 10.62, 6.46, 0.956, 1.25, 5.24, 5.39 |
| 5MWs | 16.24, 7.09, 0.938, 1.01, 5.06, 5.16 | 12.30, 6.64, 0.957, 1.06, 4.75, 4.87 | 9.72, 5.13, 0.974, 1.02, 4.61, 4.72 | 10.89, 6.56, 0.956, 1.03, 4.66, 4.78 |
| 1000T | 15.05, 7.30, 0.934, 0.79, 6.05, 6.10 | 10.39, 6.73, 0.954, 0.63, 5.64, 5.67 | 8.12, 5.45, 0.971, 0.62, 5.46, 5.50 | 10.00, 6.67, 0.953, 0.64, 5.53, 5.57 |
| 4MWsT | 15.56, 7.22, 0.938, 0.91, 5.67, 5.74 | 11.56, 6.76, 0.957, 0.79, 5.23, 5.29 | 9.26, **4.94**, 0.975, 0.77, 5.10, 5.16 | 10.54, **6.36**, 0.958, 0.79, 5.16, 5.22 |
| 5MWsT | 16.93, **7.05**, 0.940, 0.69, 4.99, **5.03** | 12.76, 6.82, 0.956, 0.60, 4.68, **4.72** | 10.11, 4.99, 0.976, 0.59, 4.53, **4.57** | 11.20, 6.42, 0.958, 0.60, 4.59, **4.63** |
| **FTIR-ECC at 12-23 km** | | | | |
| 1000 | 17.30, 5.76, 0.914, 0.81, 2.08, 2.24 | 16.01, **4.54**, 0.946, 0.75, 1.75, 1.91 | 13.23, **4.70**, 0.959, 0.76, 1.71, 1.88 | 14.59, 5.23, 0.944, 0.76, 1.74, 1.91 |
| 4MWs | 17.91, 5.58, 0.928, 0.50, 1.77, 1.84 | 15.99, 4.86, 0.943, 0.47, 1.53, 1.60 | 13.41, 4.82, 0.961, 0.47, 1.53, 1.61 | 14.79, 5.29, 0.947, 0.47, 1.55, 1.63 |
| 5MWs | 17.89, **5.43**, 0.939, 0.52, 1.65, **1.73** | 16.03, 4.94, 0.943, 0.59, 1.47, **1.59** | 13.72, 4.81, 0.962, 0.59, 1.47, **1.59** | 14.91, 5.21, 0.951, 0.58, 1.48, **1.60** |
| 1000T | 16.83, 5.64, 0.921, 0.66, 1.95, 2.06 | 15.19, 4.59, 0.945, 0.62, 1.63, 1.75 | 12.13, 4.75, 0.961, 0.62, 1.63, 1.74 | 13.72, 5.27, 0.946, 0.62, 1.65, 1.77 |
| 4MWsT | 17.25, 5.61, 0.930, 0.48, 1.82, 1.88 | 16.44, 4.65, 0.947, 0.45, 1.58, 1.65 | 14.18, 4.73, 0.963, 0.44, 1.60, 1.66 | 15.30, 5.12, 0.951, 0.45, 1.62, 1.68 |
| 5MWsT | 17.48, 5.72, 0.935, 0.38, 1.76, 1.81 | 16.91, 4.67, 0.948, 0.38, 1.56, 1.60 | 15.07, 4.77, 0.963, 0.37, 1.56, 1.61 | 16.02, **5.11**, 0.953, 0.37, 1.58, 1.62 |
| **FTIR-ECC at 22-29 km** | | | | |
| 1000 | 15.92, 3.71, 0.820, 2.00, 2.87, 3.51 | 16.23, **2.87**, 0.888, 2.17, 2.58, 3.38 | 13.94, 3.17, 0.756, 2.08, 2.56, 3.30 | 14.76, 3.50, 0.779, 2.08, 2.59, 3.33 |
| 4MWs | 16.09, 4.01, 0.793, 2.32, 2.71, 3.59 | 16.67, 2.88, 0.888, 2.66, 2.47, 3.63 | 14.16, 3.21, 0.767, 2.54, 2.53, 3.59 | 14.80, 3.64, 0.773, 2.53, 2.53, 3.59 |
| 5MWs | 16.00, 4.10, 0.783, 2.55, 2.78, 3.79 | 16.43, 2.87, 0.892, 2.91, 2.55, 3.88 | 14.06, 3.23, 0.777, 2.80, 2.62, 3.84 | 14.79, 3.68, 0.776, 2.79, 2.63, 3.84 |
| 1000T | 16.80, **3.54**, 0.866, 0.88, 3.10, 3.22 | 15.56, 3.12, 0.856, 0.78, 2.80, 2.91 | 13.88, 3.22, 0.735, 0.76, 2.77, 2.89 | 14.51, 3.55, 0.778, 0.77, 2.80, 2.90 |
| 4MWsT | 16.28, 3.56, 0.864, 0.77, 3.01, 3.11 | 15.61, 3.36, 0.844, 0.67, 2.63, 2.71 | 14.45, **3.05**, 0.774, 0.67, 2.69, 2.77 | 14.96, **3.43**, 0.801, 0.68, 2.71, 2.79 |
| 5MWsT | 16.31, 3.67, 0.856, 0.74, 2.94, **3.03** | 15.28, 3.53, 0.829, 0.66, 2.59, **2.67** | 14.47, 3.12, 0.775, 0.65, 2.65, **2.73** | 15.06, 3.50, 0.798, 0.66, 2.67, **2.75** |

attributed to the temperature fit. The same pattern is documented for the tropospheric O$_3$ concentrations even though the differences among retrieval strategies are not as significant as at higher altitudes. Note that these scatter values agree well with the expected uncertainty for ECC sondes ($\sim$5–15%) and with the FTIR theoretical error estimation (recall Section 3.2.2), as

well as with previous works (Schneider et al., 2008b; García et al., 2012; Duflot et al., 2017, and references therein). As stated in these studies, the limited vertical sensitivity of the FTIR profiles could account for part of the dispersion observed between both datasets. Other sources of discrepancies might be the different observing geometries (i.e. the two measurement techniques sample different air masses).

The vertical stratification observed in the median RD profiles also differs between both FTIR instruments (Figure 7 (a), (d),

and (g)). While all set-ups consistently show larger biases up to the UTLS region for the IFS 120M, the differences between the instruments are minimised beyond the middle stratosphere. The median bias varies from $\sim$17% to $\sim$10% at 5 km for the 5MWs/5MWsT in the 1999-2004 and 2008-2018 periods, respectively, while for the same configurations the bias is $\sim$10% at 29 km for both periods.

In summary, considering the integrated PCs (less dependent on the FTIR vertical sensitivity) and the 2008-2018 period as ref-

erence (better instrumental alignment and more FTIR-ECC coincidences), the best overall performance is documented for the set-ups using narrow micro-windows and simultaneous temperature fits in the troposphere and stratosphere regions (Table 3). In the UTLS altitudes, where O$_3$ is particularly variable, the broad micro-window strategy seems to provide the best agreement with respect to ECC data. Nevertheless, the differences among strategies lie within the respective error confidence intervals, thereby no robust conclusions can be reached. In addition, it is fair to admit that the ECC comparison only allows the FTIR

vertical profiles to be analysed in detail up to $\sim$30 km. However, compensations among the ILS, temperature, measurement errors, and O$_3$ vertical distribution should occur at higher altitudes, which leads to the usage of narrow micro-windows (and temperature fit) clearly providing the best results in the integrated total columns, as documented by the FTIR-Brewer comparison. Unfortunately, ECC sondes do not usually reach altitudes higher than 30-34 km, thereby other measurement techniques, such as microwave or LIDAR O$_3$ profiles, would be of great use for further completing the quality assessment.

**5    Summary and Conclusions**

Accurate ozone (O$_3$) products are mandatory to monitor the evolution of the Earth's atmosphere system. In this context, the current paper has assessed the effect of using different retrieval strategies on the quality of O$_3$ products from ground-based FTIR spectrometry, with the aim of providing an improved O$_3$ retrieval that could be applied at any NDACC FTIR station. For this purpose, the high-quality NDACC FTIR measurements taken at the subtropical Izaña Observatory (IZO) between 1999

and 2018 has been utilised. The 20-year time series of O$_3$ observations has allowed us to assess, on the one hand, the quality and long-term consistency of the different FTIR O$_3$ products and, on the other hand, to evaluate the influence of instrumental status on the O$_3$ retrievals.

Quality of the FTIR O$_3$ products improves as the retrieval strategies become more refined by considering O$_3$ absorption lines in specific narrow micro-windows (between 991 and 1014 cm$^{-1}$) instead of using the traditional broad spectral region

(between 1000 and 1005 cm$^{-1}$). Approaches using narrow micro-windows have theoretically and experimentally proven to be superior due to the their greater vertical sensitivity, smaller expected uncertainties, and better agreement with respect to independent data. The optimal selection of the spectral $O_3$ micro-windows can enhance the precision of FTIR $O_3$ TCs by $\sim$0.1-0.2% with respect to the coincident NDACC Brewer observations taken as reference, leading to a conservative precision of $\sim$0.6-0.7% for the FTIR products. But, at the same time, they have shown to be consistent with the standard NDACC set-up

(i.e. no important biases were found between the different retrieval strategies).

In addition, independently of the $O_3$ absorption lines used, the simultaneous atmospheric temperature retrieval has been found to be a very useful tool for $O_3$ monitoring by ground-based NDACC FTIR systems. The scatter with respect to the Brewer data is found to be reduced up to $\sim$0.2% when applying a temperature fit for those strategies also using narrow $O_3$ absorption lines. However, this improvement can only be reached provided the FTIR instrument is properly characterised and

stable over time (e.g. IFS 120/5HR spectrometer). For more unstable instruments, such as the IFS 120M, the inclusion of atmospheric temperature fit in the $O_3$ retrieval procedure may not be recommendable, since it worsens the quality of FTIR $O_3$ products due to the increase in the cross-interference with instrumental performance. The broad 1000 cm$^{-1}$ region seems to be the most sensitive to this effect. Another fact that strongly distinguishes the broad and narrow set-ups is the presence of strong $H_2O$ absorbing lines in the 1000 cm$^{-1}$ region, which could be critical for humid FTIR sites if the $H_2O$ cross-interference is not

properly taken into account. In this sense, using one-step or two-step retrieval strategies (retrieving $H_2O$ and $O_3$ in the same or in two separated steps, respectively) has been found to be valid and provide consistent results.

Regarding the vertical $O_3$ distribution, the important cross-interference between the $O_3$ and temperature profiles, and the instrumental status results in a differentiated performance of the set-ups depending on the altitude range. The best overall performance is documented for the set-ups using narrow micro-windows and simultaneous temperature fits in the troposphere

and stratosphere regions, while at tropopause altitudes the broad micro-window strategy seems to provide the best agreement with respect to ozonesonde data.

The effect of the most influential settings on FTIR $O_3$ retrieval procedure has been examined in this paper. Nevertheless, there is great potential for further improving the precision and accuracy of FTIR $O_3$ products, as well as their harmonisation within the NDACC IRWG community (comprising instruments and retrieval strategies). Additional efforts could be made with

the treatment of the instrumental response, through the evaluation of the Instrumental Line Shape function in a consistent manner, given its important effects on $O_3$ retrievals. In addition, testing the proposed $O_3$ set-ups at different NDACC FTIR stations (under different humidity conditions, latitudes, altitudes, etc.) would indeed motivate the NDACC FTIR community to revise the standard $O_3$ retrieval strategy. An improved $O_3$ monitoring could help to estimate more precisely the small expected signal of recovery or decline of $O_3$ concentrations, both for integrated total columns and vertical distributions, at a global

scale. This is particularly challenging in those regions where $O_3$ concentrations are less variable, such as at the tropical and subtropical latitudes. Furthermore, new opportunities would show up to better understand the different and outstanding $O_3$ impacts on Earth climate system, improving their representation in the current global climate models and, thus, the knowledge of their long-term evolution.

## Appendix A: Water Vapour Treatment

This appendix addresses the impact of the treatment of $H_2O$ on $O_3$ retrievals for all $O_3$ set-ups comparing the one-step and two-step retrieval strategies, where:

– One-step approach refers to simultaneously retrieve the $H_2O$ and $O_3$ profiles, using a Tikhonov-Philips slope constraint for both gases, so that the microwindow of 896.4–896.6 cm$^{-1}$ is added for a better $H_2O$ determination (as done at the NDACC FTIR Lauder and Wollongong sites in Vigouroux et al. (2015)).

– Two-step approach refers to the strategy followed in the current paper and explained in detail in Section 3.1, where the $H_2O$ a priori profiles, previously retrieved in dedicated $H_2O$ microwindows for each spectrum, are then scaled in the $O_3$ retrieval.

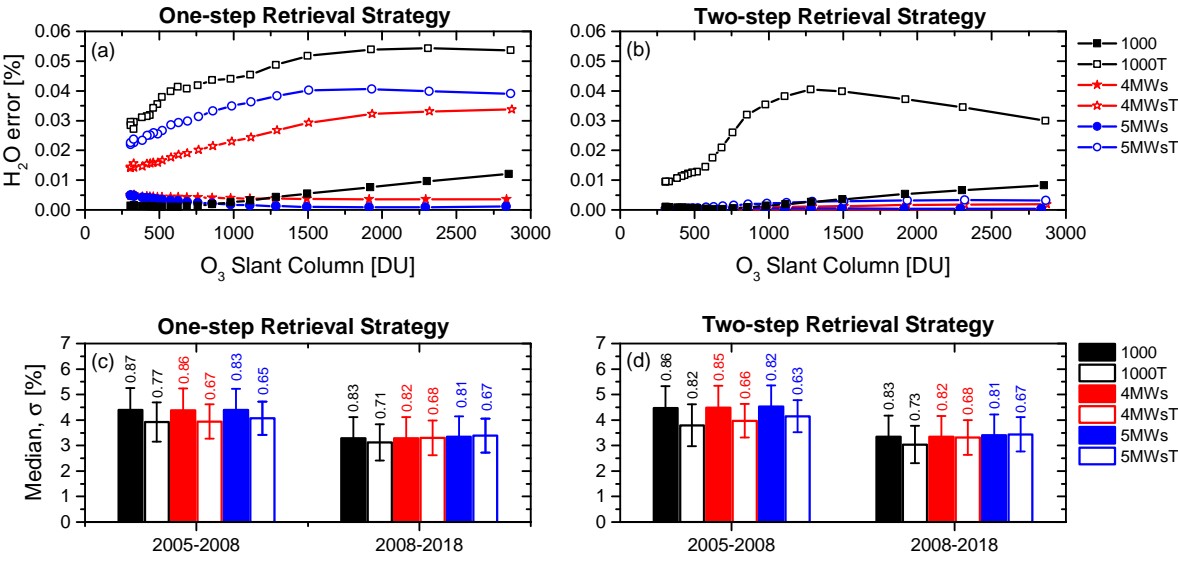

**Figure A1.** Summary of the impact of the treatment of $H_2O$ on $O_3$ retrievals for all $O_3$ set-ups. (a) and (b) $H_2O$ cross-interference error on $O_3$ total columns [%] using the one-step and two-step retrieval strategies, respectively, as a function of $O_3$ slant column [DU] for the measurements of Figure 3. (c) and (d) statistics for the FTIR-Brewer relative differences using the one-step and two-step retrieval strategies, respectively, for the periods 2005-May 2008 and June 2008-2018. Shown median (M, in %) and standard deviation values ($\sigma$, as error bars and text in %).

  Figure A1 illustrates the theoretical assessment of the $H_2O$ cross-interference for both retrieval strategies (García et al., 2014), where it can be seen that the $H_2O$ interfering error is noticeable, but not critical given the especially dry conditions at

IZO (water vapour mixing ratio less than 0.06%) (García et al., 2021). Nevertheless, it has been found that the $H_2O$ interference strongly depends on the micro-windows used for the $O_3$ retrievals (the higher impact is observed for the 1000 spectral region as

expected from Figure 2), as well as on the treatment of the atmospheric temperature profile. One-step and two-step approaches provide consistent results when the simultaneous temperature fit is not included for all set-ups and, therefore, both can be valid to correctly minimize the $H_2O$ interference. Nonetheless, provided the documented improvement of the temperature retrieval

is pursued, the two-step strategy ought to be used. In this sense, the two-step strategy drastically reduces the $H_2O$ interfering error for those set-ups using narrow micro-windows when the simultaneous temperature fit is included (4MWsT/5MWsT), leading to expected errors on the $O_3$ total columns smaller than 0.01%. The $H_2O$ interfering effect also drops for the 1000 spectral region, but to a lesser extent, given the presence of important $H_2O$ absorption lines in that region (recall Figure 2). This should be especially taken into account for FTIR stations located in humid environments.

The comparison to Brewer observations (Figure A1 (c) and (d)) also corroborates the theoretical results. It is worth highlighting the fact that the differences found between the two strategies are in excellent agreement with the estimated $H_2O$ interfering error values (Figure A1 (a) and (b)).

## Appendix B: Uncertainty Analysis

Theoretical uncertainties of FTIR products can be estimated by following the formalism detailed by Rodgers (2000), which

includes the effect of smoothing error (SE), spectral measurement noise, and different model parameter sources. The difference between the retrieved state, $\hat{x}$, and real state, $x$, can therefore be written as a linear combination of the a-priori state, $x_a$, real and estimated model parameters, $p$ and $\hat{p}$ respectively, and measurement noise $\epsilon$:

$$(\hat{x} - x) = (\mathbf{A} - \mathbf{I})(x - x_a) + \mathbf{G}\mathbf{K}_p(p - \hat{p}) + \mathbf{G}\epsilon, \tag{B1}$$

where $\mathbf{G}$ represents the gain matrix, $\mathbf{K}_p$ the sensitivity matrix to the model parameters, $\mathbf{I}$ the identify matrix, and $\mathbf{A}$ the

averaging kernel matrix.

The first term of Eq. (B1) refers to smoothing error, which has been calculated as $(\mathbf{A} - \mathbf{I})\mathbf{S}_{\mathbf{a}\mathbf{O}_3}(\mathbf{A} - \mathbf{I})^T$. The $\mathbf{S}_{\mathbf{a}\mathbf{O}_3}$ matrix is the $O_3$ a-priori covariance matrix, which has been computed in this work from the 1999-2018 ECC sonde climatology at IZO according to Schneider and Hase (2008). Note that ECC sondes usually burst between 30 and 34 km, hence this climatology was completed beyond 31 km by using the WACCM-version 6 simulations for subtropical latitudes.

The error covariance matrix for measurement noise ($\mathbf{S}_{\mathbf{x},\epsilon}$) is analytically calculated by

$$\mathbf{S}_{\mathbf{x},\epsilon} = \mathbf{G}\mathbf{S}_{\mathbf{y},\epsilon}\mathbf{G}^T, \tag{B2}$$

where $\mathbf{S}_{\mathbf{y},\epsilon}$ is the covariance matrix for measurement noise in the measurement.

The error contribution of the model parameters $p$ can be analytically estimated through the respective error covariance matrices $\mathbf{S}_{\mathbf{x},\mathbf{p}}$:

$$\mathbf{S}_{\mathbf{x},\mathbf{p}} = \mathbf{G}\mathbf{K}_{\mathbf{p}}\mathbf{S}_{\mathbf{p}}\mathbf{K}_{\mathbf{p}}^T\mathbf{G}^T, \tag{B3}$$

where $\mathbf{S}_{\mathbf{p}}$ is the covariance matrix of the uncertainties $\mathbf{\Delta p}$. In the current paper, $\mathbf{S}_{\mathbf{p}}$ is estimated considering error sources, values, and partitioning between random and systematic contributions listed in Table B1. They have been identified as the

typical error sources and values affecting the different FTIR products (e.g. Hase, 2007; Schneider and Hase, 2008; García et al., 2016; Gordon et al., 2022, and references therein). The statistical and systematic contributions of total parameter errors (TPE, displayed in Figure 3) are then calculated as the square root sum of the squares of all statistical and systematic errors considered, respectively. Note that measurement noise is considered as purely random, while spectroscopic parameters are purely systematic.

**Table B1.** Error sources and assumed values used in the theoretical uncertainty estimation. Last column shows the contribution of statistical (ST) and systematic (SY) sources to total error. Chann.: Channeling; MEA: Modulation Efficiency Amplitude; PE: Phase Error; Int.: Intensity; $\nu$-scale: Spectral position; S: Intensity parameter; $\nu$: Pressure broadening parameter.

| Error Source | Error | ST/SY |
|---|---|---|
| Baseline (Chann. and Offset) | 0.1% and 0.1% | 50/50 |
| MEA and PE (ILS) | 1% and 0.01 rad | 50/50 |
| Temperature profile | 2 K (<50 km) | 70/30 |
| | 5 K (>50 km) | 70/30 |
| Line of Sight (LOS) | 0.001 rad | 90/10 |
| Solar Lines (Int. and $\nu$-scale) | 1% and $10^{-6}$ | 80/20 |
| Spectroscopy | 5% for S and 5% $\nu$ | 0/100 |

As shown in Figure B1, when considering statistical error sources, the main contributors are the atmospheric temperature profile for set-ups without temperature fit (error values between 2.0-2.5%), and possible mis-alignments of the FTIR's solar tracker, given by the LOS, for all set-ups (error values up to 1.0% at larger $O_3$ concentrations). By contrast, $O_3$ TCs are almost insensitive to errors due to ILS uncertainties and measurement noise (error values smaller than 0.1-0.2%). For the set-ups with a simultaneous temperature retrieval, this fit generates a significant cross-interference with the ILS function, leading to an increment of the ILS error contribution (Schneider and Hase, 2008; García et al., 2012), but also with the measurement noise and smoothing error (especially for 1000T set-up). However, in return, the temperature error contribution is nearly eliminated, leading to total ST budget considerably improves by $\sim$1%. It is worth highlighting that the negative cross-interference between the temperature retrieval and SE is significant for $O_3$ SCs beyond $\sim$500 DU due to the loss of FTIR vertical sensitivity as $O_3$ SCs increase (this threshold encompasses less than 20% of the $O_3$ observations at IZO). For the typical $O_3$ concentrations observed at IZO, SE is improved by the temperature fit and smaller for the 4MWsT/5MWsT set-ups. This result is corroborated by the comparison to coincident Brewer observations, as shown in Table 2.

Note that in the current work the measurement noise depends on the quality of the fitted spectra (Hase et al., 2004). Therefore, large values of measurement noise error are observed, where the fit residuals are slightly larger (especially in the broad spectral region, see Figure B1).

Table B1 lists the uncertainty values representative of the IFS 120/HR instrument in the period 2005-2008. However, the error sources associated to instrument status (i.e. ILS function, solar pointing, and measurement noise) can be different depending

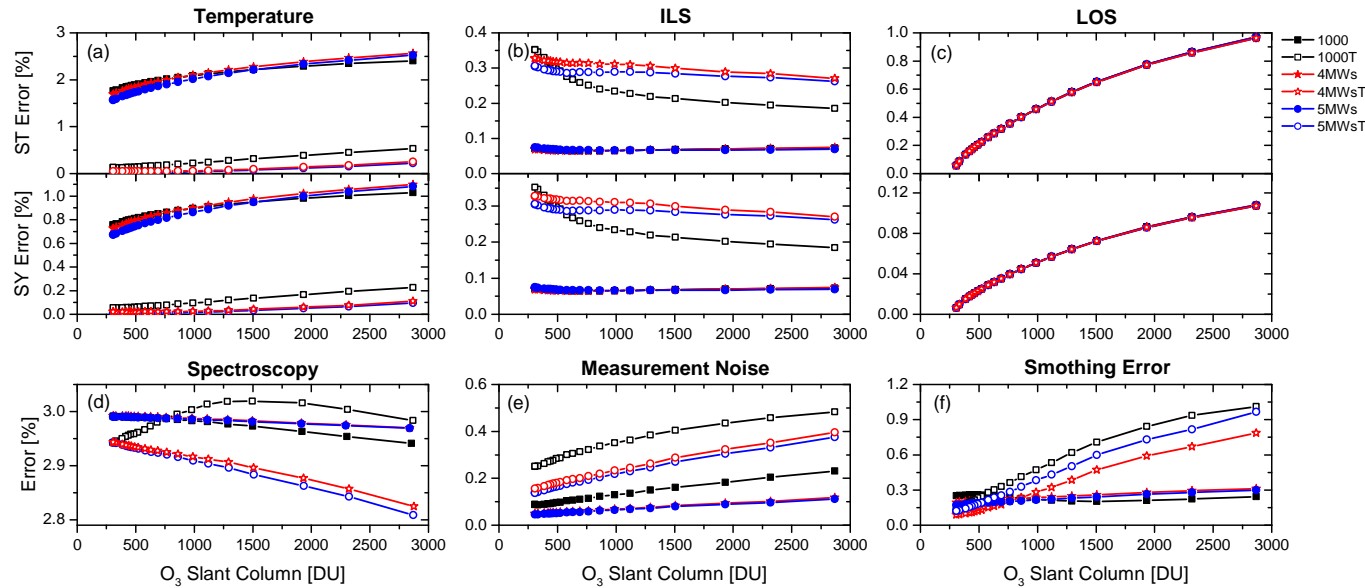

**Figure B1.** Estimated statistical (ST) and systematic (SY) errors [%] for $O_3$ TCs retrieved from all set-ups for different error sources ((a) atmospheric temperature profile, (b) ILS function, (c) LOS, (d) spectroscopic parameters, (e) measurement noise, and (f) smoothing error) as a function of $O_3$ slant column [DU] for the measurements of Figure 3.

on FTIR spectrometer's quality and stability. In order to account for this fact, Figure B2 summarises the effect of the different sets of error values on $O_3$ TCs for the measurement day of Figure B1. Note that this figure only includes the error estimations for different ILS and LOS configurations (keeping the temperature profile and measurement errors fixed), and for the set-ups 1000/1000T and 5MWs/5MWsT set-ups (the 4MWs/4MWsT error estimations are quite similar to the 5MWs/5MWsT ones, therefore they have been omitted for simplicity).

The effect of the different error sets is only noticeable when the temperature profile is simultaneously estimated with $O_3$ concentrations likely due to the interference between the ILS function and temperature retrieval. Under these conditions, statistical TPE values range from ~0.5% for uncertainties of 0.5% in the MEA and of 0.005 rad in the PE (representative of the well-aligned IFS 120/5HR instrument for the period 2008-2018) up to ~1.5% for an MEA error of 5% and PE of 0.02 rad (representative of the unstable IFS 120M spectrometer). These estimated uncertainties reproduce well the changes observed in

FTIR $O_3$ quality for the different periods when comparing to Brewer observations (see Table 2). The cross-interference between the temperature fit and other error sources is also evident for systematic contributions, especially for the worst scenario of ILS degradation (MEA uncertainty values of 5%). Note that the inconsistency for the 1000T set-up is also observed for the different ILS set errors, corroborating the findings discussed in Section 3.2.2.

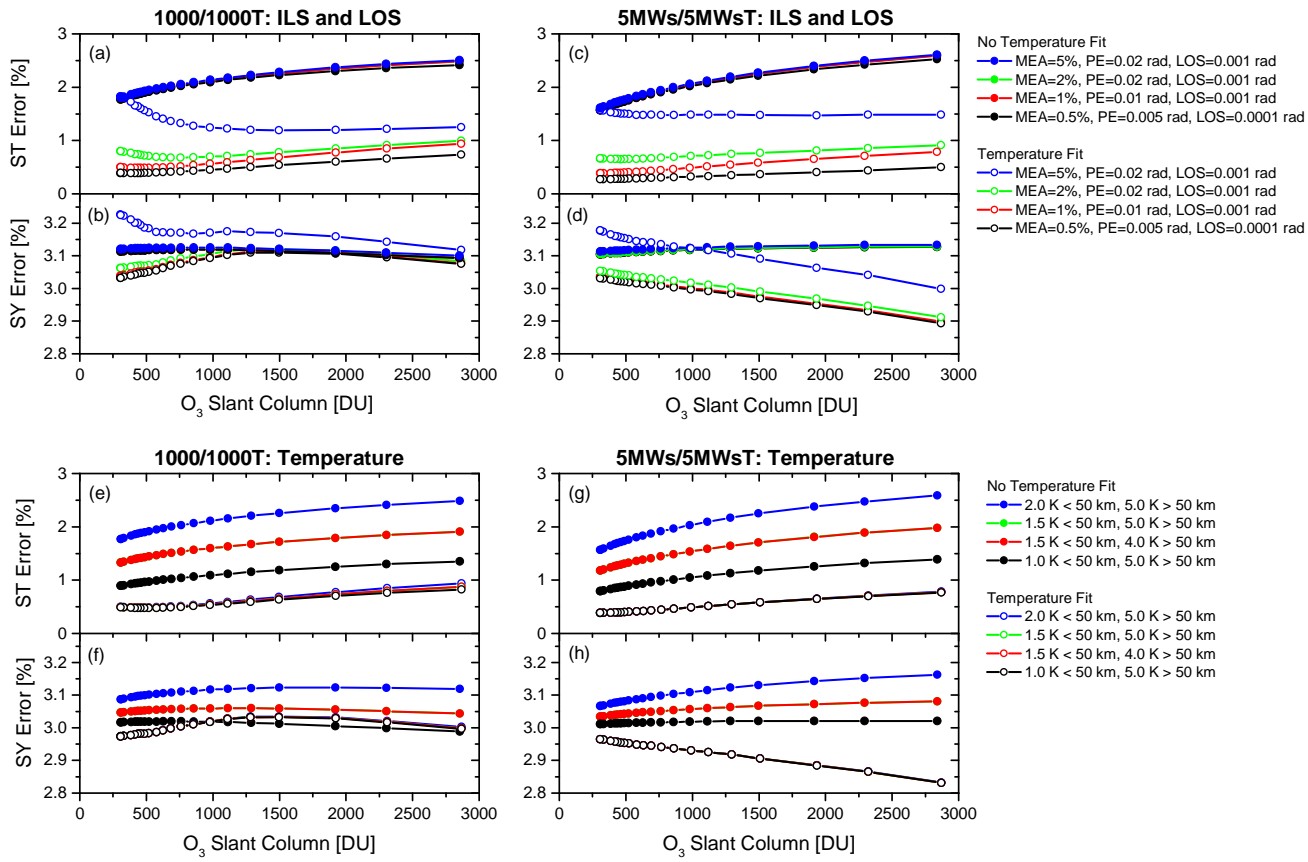

**Figure B2.** Estimated statistical (ST) and systematic (SY) TPE [%] for $O_3$ TCs retrieved from the set-ups 1000/1000T (a, b, e and f) and 5MWs/5MWsT (c, d, g and h) for different error configurations of the ILS function, LOS and atmospheric temperature profile as a function of $O_3$ slant column [DU] for the measurements of Figure 3.

*Data availability.* The FTIR and Brewer data are available by request from the corresponding authors, while the ECC ozone sondes are available from the NDACC archive (www.ndaccdemo.org/).

*Author contributions.* O.G. and E.S. designed and wrote the structure and methodology of the current paper, and computed the calculations required. M.S., F.H., and T.B. participated in the retrieval analysis. F.H. is the author of LINEFIT and PROFFIT codes. O.G., M.S. E.S., and E.S. taken the routine FTIR measurements, and performed the maintenance and quality-control of the FTIR instruments. A.R., S.F., and V.C. are responsible of maintenance and quality-control of the Brewer spectrometers, as well as of estimating the NDACC Brewer ozone observations. C.T. and N.P. are in charge of the ozone sonde programme at Izaña Observatory. Finally, all authors discussed the results and contributed to the final paper.

*Competing interests.* The authors declare no conflict of interest.

*Acknowledgements.* The Izaña FTIR station has been supported by the German Bundesministerium für Wirtschaft und Energie (BMWi) via DLRunder grants 50EE1711A and by the Helmholtz Association via the research programme ATMO. In addition, this research was

funded by the European Research Council under FP7/(2007–2013)/ERC grant agreement no. 256961 (project MUSICA), by the Deutsche Forschungsgemeinschaft for the project MOTIV (Geschäftszeichen SCHN 1126/2-1), by the Ministerio de Economía y Competitividad from Spain through the projects CGL2012-37505 (project NOVIA) and CGL2016-80688-P (project INMENSE), and by EUMETSAT under its Fellowship Programme (project VALIASI).

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
