# Peer review of "Improved ozone monitoring by ground-based FTIR spectrometry"

_Atmospheric Measurement Techniques, 2021_

## Author Comment (AC1)

**Manuscript: amt-2021-67**

Title: "Improved ozone monitoring by ground-based FTIR spectrometry" by Omaira E. García et al., Atmos. Meas. Tech. Discuss., https://doi.org/10.5194/amt-2021-67-RC1, 2021

**Response to Referee#3**

**General comments:**

The idea of homogenizing the retrieval strategy is convincing. The strategy found at IZO indeed enhances precision when comparing to Brewer data. However, the 5 different strategies do not exhibit important differences (biases) among them and the choice of the optimum strategy should be clarified. Applying the selected strategy to other NDACC measurements to verify whether this optimum strategy could be useful to the FTIR community would make the paper gain in scientific impact.

Overall, the paper is well written and structured but the abstract and conclusion sections are too vague and do not provide a concise and complete summary. These sections would need rephrasing to better highlight the main ideas/results of this work. In addition, figures would need clarity improvements. The number of figures should be reduced to fit the main scientific results.

**Specific comments:**

If the goal is to derive homogeneous O3 retrievals strategy within NDACC, why not trying the optimized strategy tested from the IZO dataset to another mid-latitude or polar NDACC measurements? This strategy is applied to IZO measurements, where, as stated in the text, is located in very dry atmospheric conditions. What happen to this optimized strategy when O3 is monitored in a much more humid environment? What would be the effect of H2O line interferences?

The authors agree with the Referee in that testing the proposed O3 set-ups on different NDACC FTIR stations (under different humidity conditions, latitudes, altitudes...) would indeed strengthen the results observed at Izaña Observatory (IZO). That might also motivate the NADCC FTIR community to revise the standard O3 retrieval strategy. In fact, discussions have already started with different NDACC stations to carry out a harmonised testing. However, this comprehensive study is not a simple matter, and requires reaching a consensus on several important factors among the participating stations, such as the treatment of water vapour interference (one-step and two-step strategies like in the current paper), ILS characterisation, retrieval code (currently two retrieval softwares are used within the FTIR community), retrieval settings (e.g. spectroscopic database), etc.

Given the importance of water vapour absorption across the infrared spectrum, the treatment of  $H_2O$  in  $O_3$  retrievals should be carefully considered in the inversion procedure. For that reason, all the  $O_3$  set-ups analysed in the current work are based on a two-step retrieval strategy, which minimises the  $H_2O$  interferences, allowing the conclusions drawn to be valid for many more humidity environments. However, the authors agree with both Referees in that the treatment of  $H_2O$  and its potential interferences are an important topic and can be treated in greater depth in the paper, leading to more robust conclusions. Accordingly, new information will be added to the revised manuscript as follows:

On the one hand, Figure 2 of the preprint has been modified by including the changes in the FTIR radiances for the spectral micro-windows used for the  $O_3$  retrievals due to different changes in the H2O content: 50% (total column of 16.1 mm), 100% (total column of 21.5 mm), and 200% (total column of 32.3 mm) (the actual total column is 10.7 mm). These values could account for typical H2O content and variations at sites with greater humidity (see Figure 1 below). As observed, the spectral signatures of H2O variations are much stronger in the broad 1000 spectral region than in the narrow micro-windows (4MWs/5MWs), indicating that the quality of the O3 products in that region strongly depends on a correct interpretation of the spectroscopic H2O interferences.

Figure 1: (a) Spectral micro-windows used in the different FTIR O3 retrieval strategies: broad window used in the set-ups 1000/1000T, encompassing the 1000-1005 cm-1 spectral region, and the four and five micro-windows used in the set-ups 4MWs/4MWsT, between ~991 and 1009 cm-1, and in 5MWs/5MWsT between ~991 and 1014 cm-1, respectively. (b) Spectral changes in the FTIR radiances ( $\Delta R$ ) due to changes in H2O content of 50% (total column of 16.1 mm), 100% (total column of 21.5 mm), and 200% (total column of 32.3 mm).The actual H2O total column is 10.7 mm.

On the other hand, the impact of the treatment of  $H_2O$  on  $O_3$  retrievals for all  $O_3$  set-ups will be addressed in a dedicated Appendix, using the one-step and two-step retrieval strategies (following Referee#2's comment), where:

- One-step refers to simultaneously retrieving the H2O and O3 profiles, using a Tikhonov-Philips slope constraint for both gases and adding the microwindow of 896.4–896.6 cm-1 for a better H2O determination (as done at the NDACC FTIR Lauder and Wollongong sites in Vigouroux et al., 2015).
- Two-step refers to the strategy followed in the current paper, where the H2O a priori profiles are only scaled in the O3 retrieval but these a priori profiles have been preliminarily retrieved in dedicated H2O microwindows for each spectrum (Schneider et al., 2012).

The new Appendix will include the theoretical assessment of  $H_2O$  cross-interference via  $H_2O$  interfering error according to García et al. (2014). As can be seen in Figure 2, the  $H_2O$  interfering error is noticeable (less than 0.06%), but not critical. Nevertheless, it has been found that the  $H_2O$  interference strongly depends on the spectral region used for the  $O_3$  retrievals (the higher impact is observed for the 1000 spectral region as expected from Figure 1), as well as on the treatment of the atmospheric temperature profile (with or without simultaneous retrieval). Note that the twostep strategy drastically reduces the H2O interfering error for those set-ups using narrow microwindows when the simultaneous temperature fit is included (4MWsT and 5MWsT set-ups), leading to expected errors on the O3 total columns smaller than 0.01%. The H2O interfering effect also drops for the 1000 spectral region, but to a lesser extent given the presence of important H2O absorption lines in that region (see Figure 1 above). These results confirm that using narrow O3 absorption lines, along with a two-step inversion strategy to estimate the H2O profile in a dedicated H2O profile fit prior to the O3 retrievals result in a superior O3 strategy.

Figure 2.  $H_2O$  interfering error on  $O_3$  total columns (in %) for all  $O_3$  set-ups using the one-step and twostep retrieval strategies for the exemplary day used in the paper (31st August 2007).

Additionally, the new Appendix will include the comparison of Brewer observations to FTIR O3 total columns from the different O3 set-ups using one-step and two-step retrieval strategies. For the IFS 120M instrument, the spectral region used for O3 retrievals was measured with two different filters at IZO: SI between 925.30-1379.71 cm-1 and SK between 700.00-1079.71 cm-1. The results presented in the preprint (two-step strategy) were evaluated from measured SI spectra given their higher signal-to-noise ratio. Unfortunately, these spectra do not cover the 896.4–896.6 cm-1 line needed for the H2O estimation in the one-step strategy, therefore the performance of both strategies has been evaluated here using the measured SK spectra for the 120M period (1999-2004).

As summarised in Table 1, the preliminary  $H_2O$  retrievals slightly enhance the quality of the  $O_3$  retrievals with respect to the one-step strategy for all periods and set-ups. The more unstable the instrument, the greater the effect. It is worth highlighting the fact that the differences found between the two strategies are in excellent agreement with the estimated  $H_2O$  interfering error values (see Figure 2 above). Note that Table 1 also includes the comparison results for the SI spectra using the two-step retrieval strategy, corroborating the best performance of these spectra for FTIR  $O_3$  retrievals.

|                                                       | 1999-2004             | 2005-2008         | 2008-2018             | 1999-2018              |  |  |  |
|-------------------------------------------------------|-----------------------|-------------------|-----------------------|------------------------|--|--|--|
| set-up                                                | M[%], $\sigma$ [%], R | M[%], σ[%], R     | M[%], $\sigma$ [%], R | $M[\%], \sigma[\%], R$ |  |  |  |
| One-Step Retrieval Strategy (Filter SK for 1999-2004) |                       |                   |                       |                        |  |  |  |
| 1000                                                  | 4.13, 2.52, 0.871     | 4.39, 0.87, 0.971 | 3.29, 0.83, 0.982     | 3.38, 1.06, 0.971      |  |  |  |
| 4MWs                                                  | 4.04, 2.50, 0.873     | 4.38, 0.86, 0.972 | 3.29, 0.82, 0.982     | 3.37, 1.05, 0.971      |  |  |  |
| 5MWs                                                  | 4.07, 2.44, 0.879     | 4.40, 0.83, 0.974 | 3.34, 0.81, 0.983     | 3.42, 1.02, 0.973      |  |  |  |
| 1000T                                                 | 3.73, 2.21, 0.908     | 3.92, 0.77, 0.976 | 3.12, 0.71, 0.987     | 3.16, 0.89, 0.979      |  |  |  |
| 4MWsT                                                 | 3.83, 2.16, 0.917     | 3.94, 0.67, 0.982 | 3.30, 0.68, 0.988     | 3.36, 0.84, 0.982      |  |  |  |
| 5MWsT                                                 | 3.92, 2.08, 0.922     | 4.07, 0.65, 0.983 | 3.39, 0.67, 0.988     | 3.45, 0.83, 0.982      |  |  |  |
| Two-Step Retrieval Strategy (Filter SK for 1999-2004) |                       |                   |                       |                        |  |  |  |
| 1000                                                  | 3.98, 2.46, 0.874     | 4.47, 0.86, 0.970 | 3.35, 0.83, 0.982     | 3.45, 1.06, 0.970      |  |  |  |
| 4MWs                                                  | 3.92, 2.45, 0.875     | 4.49, 0.85, 0.971 | 3.34, 0.82, 0.982     | 3.43, 1.05, 0.971      |  |  |  |
| 5MWs                                                  | 4.03, 2.38, 0.882     | 4.53, 0.82, 0.973 | 3.41, 0.81, 0.983     | 3.49, 1.02, 0.972      |  |  |  |
| 1000T                                                 | 3.56, 2.21, 0.906     | 3.79, 0.82, 0.972 | 3.04, 0.73, 0.986     | 3.09, 0.90, 0.978      |  |  |  |
| 4MWsT                                                 | 3.79, 2.14, 0.916     | 3.97, 0.66, 0.981 | 3.32, 0.68, 0.988     | 3.39, 0.84, 0.981      |  |  |  |

---

## Author Comment (AC2)

Manuscript: amt-2021-67

Title: "Improved ozone monitoring by ground-based FTIR spectrometry" by Omaira E. García et al., Atmos. Meas. Tech. Discuss., https://doi.org/10.5194/amt-2021-67-RC2, 2021

**Response to Referee#2 (Corinne Vigouroux)**

*General comments*

*The study of García et al. (2021) examines the performance of different O3 retrieval strategies from FTIR (Fourier Transform InfraRed) spectrometry at the subtropical Izaña site. In particular, it studies the effect of the spectral region used for O3 retrievals and of the inclusion of an atmospheric temperature profile fit, which is of high interest for the whole IRWG (Infra-Red Working Group) of NDACC (Network for the Detection of Atmospheric Composition Change) that aims at providing the best possible O3 product. The quality assessment of the different FTIR O3 products (total columns and profiles) is carefully led, both theoretically and experimentally by comparing with Brewer and sondes coincident measurements. Therefore, I recommend the publication of this paper in AMT, after a few comments/suggestions and questions (listed below) are addressed.*

*Specific comments:*

*- **Section Introduction, l. 47** "However, others are still flexible and station-dependent (e.g. the inclusion of a temperature retrieval…"*

*The temperature retrieval is kept as an alternative in the document IRWG (2014), but in practice, in the NDACC archive, all sites are consistent in not doing the temperature retrieval. It would be better to specify this to not let the NDACC users think that the NDACC products are not harmonized in the choice of retrieval settings. Therefore, the current NDACC homogenization is in very good shape; mainly the ILS treatment is not harmonized. And the impact of different ILS treatment is not treated in the current work. Therefore, I would correct l.47 (and followings) accordingly.*

*Also, in l. 50 "..much efforts should be paid…" should be replaced by "… additional efforts could …" Although this additional effort is mainly related to ILS, and not treated here, so this statement could also be put in the conclusions as a perspective to be done in the IRWG, rather than an introduction to the present work.*

*However, even if the retrieval settings are well harmonized, this does not mean that they cannot be improved. So I would more emphasize the present work to be a research towards a better strategy (micro-windows; temperature retrievals) to be proposed – if proven better - to the IRWG in replacement of the current harmonized one (this presentation of the study is actually well done in the conclusions). To my opinion the present work is not improving the harmonization and the network consistency, but is pushing towards an improvement of the retrieval strategy itself (which is very valuable).*

*For this exercise (finding a better strategy than the current IRWG one) it would have been good to include a few more stations to prove that the conclusions at Izaña are valid at other sites as well.*

The statements related to the harmonization of the NDACC IRWG O3 products will be made clearer, putting the study into a better context in accordance with the Referee's comment. Additionally, the authors agree with the Referee in that the necessity of improving the network consistency (lines 50-52) fits better as a conclusion of the current work. Accordingly, it will be moved to the Conclusion section.

The authors agree with the Referee in that testing the proposed $O_3$ set-ups on different NDACC FTIR stations (under different humidity conditions, latitudes, altitudes...) would indeed strengthen the results found at Izaña Observatory (IZO). That might also motivate the NADCC FTIR community to revise the standard $O_3$ retrieval strategy. In fact, discussions have already started with different NDACC stations to carry out a harmonised testing. However, this comprehensive study is not a simple matter, and requires reaching a consensus on several important factors among the participating stations, such as the treatment of water vapour interference (one-step and two-step strategies like in the current paper), ILS characterisation, retrieval code (currently two retrieval softwares are used within the FTIR community), retrieval settings (e.g. version of spectroscopic database), etc.

Therefore, the authors agreeing on the harmonisation study would be very useful indeed, but we consider it to be such a huge exercise that it should be addressed in two separated works: the first one addressing the comprehensive study performed in the current study (theoretical and experimental quality assessment); and a second work, where the lessons learnt from the first study can be easily applied at different NDACC stations under different casuistry. This reflection will be included in the conclusions of the revised manuscript.

*- **Section 2.2, Brewer and ECC sondes**: when you give the uncertainty for Brewer and sondes, is it the random, systematic or total one? It should be specified for the interpretation of the comparisons with FTIR (bias, standard deviation).*

For the RBCC-E Brewer instruments the uncertainty values correspond to the total uncertainty (standard uncertainty, k=1). However, there is a mistake in the values reported in the original manuscript, according to Gröbner et al. (2017). The overall uncertainty of about 1 Dobson Unit corresponds to the $O_3$ total columns (TCs) as observed by the reference QUASUME spectrometer, not by the RBCC-E Brewer instruments. For the latter the standard uncertainty (k=1) is estimated to be between 1.2% and 1.5% for the $O_3$ TCs. This statement will be corrected in the revised text accordingly.

Regarding the ECC sondes, the values reported in the manuscript correspond to the total uncertainties, which were theoretically estimated as a composite of the contributions of the individual uncertainties of the different instrumental parameters (i.e. measured sensor current, background current, conversion efficiency, temperature of the gas sampling pump and volumetric flow rate) (WMO, 2014). As is also documented by the WMO reference report, these theoretical estimates match the precision and accuracy obtained from multiple experimental intercomparisons available in literature.

*- **Section 3.1 Ozone retrieval strategies**:*

*- **l.159: H2O treatment**: did you test to simultaneously retrieve the H2O profile in a one-step approach (as done at Lauder/ Wollongong in Vigouroux et al. 2015)? The results might be equivalent to your 2 step approach, while being more simple.*

Given the importance of water vapour absorption across the infrared spectrum, the treatment of $H_2O$ in $O_3$ retrievals should be carefully considered in the inversion procedure. For that reason, all the $O_3$ set-ups analysed in the current work are based on a two-step retrieval strategy, which minimises the $H_2O$ interferences, allowing the conclusions drawn to be valid for many more humidity environments. However, the authors agree with both Referees in that the treatment of $H_2O$ and its potential interferences are an important topic and can be treated in greater depth in the paper, leading to more robust conclusions. Accordingly, new information will be added to the revised manuscript as follows:

On the one hand, Figure 2 of the preprint has been modified by including the changes in the FTIR radiances for the spectral micro-windows used for the $O_3$ retrievals due to different changes in the $H_2O$ content: 50% (total column of 16.1 mm), 100% (total column of 21.5 mm), and 200% (total column of 32.3 mm) (the actual total column is 10.7 mm). These values could account for typical $H_2O$ content and variations at sites with greater humidity (see Figure 1 below). As observed, the spectral signatures of $H_2O$ variations are much stronger in the broad 1000 spectral region than in the narrow micro-windows (4MWs/5MWs), indicating that the quality of the $O_3$ products in that region strongly depends on a correct interpretation of the spectroscopic $H_2O$ interferences.

[Figure]

Figure 1. (a) Spectral micro-windows used in the different FTIR $O_3$ retrieval strategies: broad window used in the set-ups 1000/1000T, encompassing the 1000-1005 cm$^{-1}$ spectral region, and the four and five micro-windows used in the set-ups 4MWs/4MWsT, between ~991 and 1009 cm$^{-1}$, and in 5MWs/5MWsT between ~991 and 1014 cm$^{-1}$, respectively. (b) Spectral changes in the FTIR radiances ($\Delta$R) due to changes in $H_2O$ content of 50% (total column of 16.1 mm), 100% (total column of 21.5 mm), and 200% (total column of 32.3 mm).The actual $H_2O$ total column is 10.7 mm.

On the other hand, the impact of the treatment of $H_2O$ on $O_3$ retrievals for all $O_3$ set-ups will be addressed in a dedicated Appendix, using the one-step and two-step retrieval strategies (following Referee#2's comment), where:

1. One-step refers to simultaneously retrieving the $H_2O$ and $O_3$ profiles, using a Tikhonov-Philips slope constraint for both gases and adding the microwindow of 896.4–896.6 cm$^{-1}$ for a better $H_2O$ determination (as done at the NDACC FTIR Lauder and Wollongong sites in Vigouroux et al., 2015).

2. Two-step refers to the strategy followed in the current paper, where the $H_2O$ a priori profiles are only scaled in the $O_3$ retrieval but these a priori profiles have been

preliminarily retrieved in dedicated $H_2O$ microwindows for each spectrum (Schneider et al., 2012).

The new Appendix will include the theoretical assessment of $H_2O$ cross-interference via $H_2O$ interfering error according to García et al. (2014). As can be seen in Figure 2, the $H_2O$ interfering error is noticeable (less than 0.06%), but not critical. Nevertheless, it has been found that the $H_2O$ interference strongly depends on the spectral region used for the $O_3$ retrievals (the higher impact is observed for the 1000 spectral region as expected from Figure 1), as well as on the treatment of the atmospheric temperature profile (with or without simultaneous retrieval). Note that the two-step strategy drastically reduces the $H_2O$ interfering error for those set-ups using narrow micro-windows when the simultaneous temperature fit is included (4MWsT and 5MWsT set-ups), leading to expected errors on the $O_3$ total columns smaller than 0.01%. The $H_2O$ interfering effect also drops for the 1000 spectral region, but to a lesser extent given the presence of important $H_2O$ absorption lines in that region (see Figure 1 above). These results confirm that using narrow $O_3$ absorption lines, along with a two-step inversion strategy to estimate the $H_2O$ profile in a dedicated $H_2O$ profile fit prior to the $O_3$ retrievals result in a superior $O_3$ strategy.

[Figure]

Figure 2. $H_2O$ interfering error on $O_3$ total columns (in %) for all $O_3$ set-ups using the one-step and two-step retrieval strategies for the exemplary day used in the paper (31st August 2007).

Additionally, the new Appendix will include the comparison of Brewer observations to FTIR $O_3$ total columns from the different $O_3$ set-ups using one-step and two-step retrieval strategies. For the IFS 120M instrument, the spectral region used for $O_3$ retrievals was measured with two different filters at IZO: SI between 925.30-1379.71 $cm^{-1}$ and SK between 700.00-1079.71 $cm^{-1}$. The results presented in the preprint (two-step strategy) were evaluated from measured SI spectra given their higher signal-to-noise ratio. Unfortunately, these spectra do not cover the 896.4–896.6 $cm^{-1}$ line needed for the $H_2O$ estimation in the one-step strategy, therefore the performance of both strategies has been evaluated here using the measured SK spectra for the 120M period (1999-2004).

As summarised in Table 1, the preliminary $H_2O$ retrievals slightly enhance the quality of the $O_3$ retrievals with respect to the one-step strategy for all periods and set-ups. The more unstable the instrument, the greater the effect. It is worth highlighting the fact that the differences found

between the two strategies are in excellent agreement with the estimated $H_2O$ interfering error values (see Figure 2 above). Note that Table 1 also includes the comparison results for the SI spectra using the two-step retrieval strategy, corroborating the best performance of these spectra for FTIR $O_3$ retrievals.

| set-up | 1999-2004 M[%], $\sigma$[%], R | 2005-2008 M[%], $\sigma$[%], R | 2008-2018 M[%], $\sigma$[%], R | 1999-2018 M[%], $\sigma$[%], R |
|---|---|---|---|---|
| | One-Step Retrieval Strategy (Filter SK for 1999-2004) | | | |
| 1000 | 4.13, 2.52, 0.871 | 4.39, 0.87, 0.971 | 3.29, 0.83, 0.982 | 3.38, 1.06, 0.971 |
| 4MWs | 4.04, 2.50, 0.873 | 4.38, 0.86, 0.972 | 3.29, 0.82, 0.982 | 3.37, 1.05, 0.971 |
| 5MWs | 4.07, 2.44, 0.879 | 4.40, 0.83, 0.974 | 3.34, 0.81, 0.983 | 3.42, 1.02, 0.973 |
| 1000T | 3.73, 2.21, 0.908 | 3.92, 0.77, 0.976 | 3.12, 0.71, 0.987 | 3.16, 0.89, 0.979 |
| 4MWsT | 3.83, 2.16, 0.917 | 3.94, 0.67, 0.982 | 3.30, 0.68, 0.988 | 3.36, 0.84, 0.982 |
| 5MWsT | 3.92, 2.08, 0.922 | 4.07, 0.65, 0.983 | 3.39, 0.67, 0.988 | 3.45, 0.83, 0.982 |
| | Two-Step Retrieval Strategy (Filter SK for 1999-2004) | | | |
| 1000 | 3.98, 2.46, 0.874 | 4.47, 0.86, 0.970 | 3.35, 0.83, 0.982 | 3.45, 1.06, 0.970 |
| 4MWs | 3.92, 2.45, 0.875 | 4.49, 0.85, 0.971 | 3.34, 0.82, 0.982 | 3.43, 1.05, 0.971 |
| 5MWs | 4.03, 2.38, 0.882 | 4.53, 0.82, 0.973 | 3.41, 0.81, 0.983 | 3.49, 1.02, 0.972 |
| 1000T | 3.56, 2.21, 0.906 | 3.79, 0.82, 0.972 | 3.04, 0.73, 0.986 | 3.09, 0.90, 0.978 |
| 4MWsT | 3.79, 2.14, 0.916 | 3.97, 0.66, 0.981 | 3.32, 0.68, 0.988 | 3.39, 0.84, 0.981 |
| 5MWsT | 3.93, 2.06, 0.922 | 4.15, 0.63, 0.983 | 3.44, 0.67, 0.988 | 3.51, 0.83, 0.982 |
| | Two-Step Retrieval Strategy (Filter SI for 1999-2004) | | | |
| 1000 | 4.29, 1.38, 0.957 | 4.47, 0.86, 0.970 | 3.35, 0.83, 0.982 | 3.46, 0.95, 0.975 |
| 4MWs | 4.28, 1.36, 0.959 | 4.49, 0.85, 0.971 | 3.34, 0.82, 0.982 | 3.45, 0.93, 0.976 |
| 5MWs | 4.35, 1.32, 0.962 | 4.53, 0.82, 0.973 | 3.41, 0.81, 0.983 | 3.50, 0.91, 0.977 |
| 1000T | 4.83, 1.97, 0.926 | 3.79, 0.82, 0.972 | 3.04, 0.73, 0.986 | 3.12, 0.90, 0.977 |
| 4MWsT | 4.84, 1.90, 0.934 | 3.97, 0.66, 0.981 | 3.32, 0.68, 0.988 | 3.40, 0.83, 0.981 |
| 5MWsT | 4.81, 1.82, 0.940 | 4.15, 0.63, 0.983 | 3.44, 0.67, 0.988 | 3.53, 0.81, 0.982 |

Table 1. Summary of statistics for FTIR-Brewer comparison for the set-ups 1000/1000T, 4MWs/4MWsT, and 5MWs/5MWsT: median (M, in %) and standard deviation ($\sigma$, in %) of the relative differences, and Pearson correlation coefficient (R) of the direct comparison for the periods 1999-2004, 2005-May 2008 and June 2008-2018, and for the entire time series (1999-2018), considering the one-step and two-step strategies for the $H_2O$ estimation.

*- I.165: ILS treatment*

*In your retrievals, the ILS is fixed to the results obtained by LINEFIT using the N2O cell-measurements. Did you try to retrieve it? (starting from LINEFIT results as a priori values), in order to e.g. improve the comparisons in the 1999-2008 periods. The LINEFIT results are also obtained with some uncertainty, and averaging kernels that do not have a full sensitivity for the*

*whole OPD. It would be interesting to see if the results of your quality assessment (by comparing with Brewer and sondes) could be improved by fitting the ILS. This would also be an interesting result for the whole IRWG and the harmonized strategy. Could you add this test for own of your set-up (e.g. 4MWS)?*

Indeed, as the Referee points out here and in her first specific comment, one of the factors that are most dependent on each NDACC FTIR site is the treatment of the spectrometer response through the Instrumental Line Shape (ILS) function (e.g. Vigouroux et al., 2008; Vigouroux et al., 2015). A precise knowledge of the ILS is essential to properly characterise the instrument performance, since the ILS affects the absorption line shape on which the retrieved information is based on. This is of special relevance for stratospheric gases, such as ozone, since the full width at the half maximum of their sharp absorption lines and of ILS have similar magnitudes (Takele Kenea et al., 2013; García et al., 2014b; Sun et al., 2018).

As part of the effort to improve the standard retrieval strategy within the NDACC IRWG community, the impact on $O_3$ FTIR retrievals of several approaches used to characterise the ILS function is being also tested in parallel with the current work. Some preliminary results were presented in the last joint NDACC-IRWG/TCCON meeting, held online in June 2021 (García et al., 2021). For example, Figure 3 shows the median and standard deviation of the relative difference between $O_3$ total columns as observed by Brewer and FTIR instruments at IZO in the three separated periods distinguished in the current work. It includes different approaches to consider the ILS function in the $O_3$ FTIR retrievals using the five-microwindow strategy proposed here (with and without simultaneous temperature retrieval) as follows:

- ✓ 5A/5AT: The ILS is monitored via independent $N_2O$-cell measurements and obtained by LINEFIT software.

- ✓ 5B/5BT: The ILS is assumed to be ideal.

- ✓ 5G/5GT: The ILS is retrieved simultaneously with $O_3$, the phase error (PE) being fitted to a constant value throughout the optical path difference (OPD) range, and the modulation efficiency amplitude (MEA) calculated by using a second-order polynomial fit of OPD.

- ✓ 5H/5HT: The $N_2O$-derived ILS is "improved" with a simultaneous fit of a PE offset along with the $O_3$ retrieval (i.e. the LINEFIT results are used as a priori values as the Referee suggests).

As can be seen in Figure 3, the joint ILS and $O_3$ retrieval allows a rough instrumental characterisation to be obtained and the precision of FTIR $O_3$ products to be slightly improved. However, the preliminary results also point out that the ILS retrieval might lead to a misinterpretation of the actual $O_3$ variations on a daily and seasonal scale.

Therefore, the authors agreeing with the Referee in that the ILS harmonisation study would be very useful, but we consider it to be such a huge exercise that it should be addressed in another separated work, where the ILS casuistry within the NDACC-IRWG community can be analysed in greater detail. The authors are already working on this study and hope to submit it for publication in the coming months.

[Figure]

Figure 3. Median (M, in %) and standard deviation (σ, in %) of the relative differences for the Brewer-FTIR comparison at IZO for the periods 1999-2004, 2005-May 2008 and June 2008-2018 for different ILS approaches (see explanation in the text).

*- I.168: temperature profiles*

*NCEP provides now 6-hourly temperature, pressure, H2O profiles. I guess that if you would use these 6-hourly profiles instead of daily means, you would decrease the effect of retrieving vs fixing your temperature profiles. And you would have a temperature covariance matrix that should have reduced values, which would decrease the uncertainties due to the fixed temperature. Why not using the best NCEP available values if it is proven that for O3 the temperature is a leading source of uncertainty?*

As shown in Figure 4, the FTIR $O_3$ measurements at IZO are mostly taken around noon. In particular, about 86% of the total observations during the 1999-2018 are concentrated in the interval 9:00-15:00 UTC, i.e., ±3 hours around the NCEP temperature and pressure profiles used as reference in the $O_3$ retrievals (12 UTC). Therefore, the 12 UTC NCEP profiles can be considered a reliable proxy of the atmospheric state at IZO for the radiative transfer calculations. Nevertheless, as the Referee suggests, greater frequent NCEP profiles might improve the overall quality of O3 retrievals, and it will be taken into consideration in the next re-evaluation of the NDACC IZO database that is expected to be carried out in 2021/2022. In this sense, a previous work analysing the effect of the intra-day variability of the pressure and temperature profiles on different FTIR products (3-hourly profiles) has shown that a mean difference and scatter of about 0.06% and 0.01%, respectively, would be expected when compared to coincident Brewer $O_3$ total column observations at IZO (García et al., 2014b).

[Figure]

Figure 4. Hourly distribution of the FTIR $O_3$ measurements taken at IZO in the period 1999-2018. The number of measurements (left axis) and the cumulative percentage (right axis) are shown.

*- Section 3.2.2 Uncertainty analysis, l. 233: "where the spectroscopic SY errors determine the total uncertainty budget (with values of ∼5%)" To my knowledge, the uncertainty due to O3 line intensity (dominating the systematic error on the total column) has been set to 3% in the IRWG (SFIT4 new release, agreement with PROFFIT users as well, B. Langerock, F. Hase, personal communication). This is quite in agreement with your Table 2 for the best measurement periods (2008-2018): bias with Brewer below 3.4%.*

*I would change this 5% value here and p. 15. L. 324.*

Following the Referee's suggestion, the uncertainty of the $O_3$ spectroscopic parameters has been set to 3% and the uncertainty estimations presented in Section 3.2.2. have been recalculated accordingly. In fact, as recently presented by Hargreaves et al. (2020), the coming version of HITRAN spectroscopic database (HITRAN 2020) will improve the $O_3$ line intensities in the 10 $\mu$m spectral band (corresponding to 1000 cm$^{-1}$) by applying a scaling factor of 3%. This improvement is based on previous analyses comparing the microwave (MW) and mid-infrared (MIR) spectral regions (10 and 5 $\mu$m), which show that the MW and mid-infrared are self-consistent but too weak, and recommended specific scaling corrections between 3-4% depending on the spectral region (MW or MIR) (Droulin et al., 2017; Birk et al., 2019; Tyuterev et al., 2019). As mentioned above, the proposed correction for $O_3$ line intensities is of 3% in the spectral region used for the FTIR $O_3$ retrievals, which agree well with the bias found between FTIR and Brewer observations and our updated uncertainty estimation. This additional information will be included in the revised manuscript.

*- l.240: smoothing error*

*If the smoothing error is getting more important when fitting the temperature, then it is important to give total error budget with smoothing included (also in Fig.5 / Table 2). To check if it's worth fitting the temperature at the end. Decision should be made using total uncertainty, smoothing included.*

As stated in Section 3.2.2 (lines 219-220), the uncertainty analysis carried out in this work follows the guidance of the NDACC IRWG (IRWG, 2014), which does not include the smoothing error contribution as part of the standard uncertainty estimations. For that reason, this error was estimated, but considered separately in the current study (see Figure 5). Nevertheless, the authors agree with the Referee in that estimating the impact of instrumental smoothing on the different $O_3$ retrieval strategies is necessary to obtain a complete view. Therefore, the total uncertainties will be included both with and without considering the smoothing contribution in the revised manuscript (Section 3.2.2).

Note that, following the Referre#3's suggestion, the Section 3.2.2. will be revised and simplified to allow for an easy read. Figure 4 of the preprint will be simplified and combined with Figure 5, and the discussion and plots showing the detailed error analysis based on the different error sources will be moved to Appendix A.

[Figure]

Figure 5. Smoothing error on $O_3$ total columns (in %) for all $O_3$ set-ups as a function of the $O_3$ slant columns (in DU) for the exemplary day used in the paper (31st August 2007).

*- Discussion p. 14 l. 296- 307:*

*It looks like the extreme RD values occur mainly during the 120M measurements period. So could it simply be that T retrieval are less stable with 120M (bad ILS), and therefore gives outliers in some of the retrievals?*

The extreme relative differences (RD) between Brewer and FTIR observations are indeed concentrated in the IFS 120M period, since the interference between an unstable instrument (bad ILS) and temperature profile is more important. Nevertheless, these extreme RD values are also observed for the IFS 120/5HR spectrometer (Figure 6 of the preprint), even during the 2008-2018 period when the FTIR instrument was properly aligned as shown in the ILS time series (Figure 1 of the preprint).

*- p.14, l.320 & p.15 Table 2:*

*It would be better for the discussion to include the total statistical error in Table 2 for different set-ups / period, and/or the root-square-sum of the precision of Brewer+ FTIR. Note that the smoothing error must be included in the total budget.*

Following the Referee's suggestion, the total statistical error has been estimated for the entire FTIR O₃ time series using the different set-ups, and included in Table 2 of the revised manuscript (see Table 2 below). Similar to the revised Section 3.2.2, total errors will be included both with and without considering the smoothing contribution for a better interpretation of the results.

Note that the root-square-sum of the Brewer and FTIR precisions can not be provided for each individual measurement as the development of standardised uncertainty assessment is currently on-going within the Brewer community.

| set-up | 1999-2004 M, $\sigma$, R, M$_{ST}$, $\sigma_{ST}$ [%], [%], - , [%], [%] | 2005-2008 M, $\sigma$, R, M$_{ST}$, $\sigma_{ST}$ [%], [%], - , [%], [%] | 2008-2018 M, $\sigma$, R, M$_{ST}$, $\sigma_{ST}$ [%], [%], - , [%], [%] | 1999-2018 M, $\sigma$, R, M$_{ST}$, $\sigma_{ST}$ [%], [%], - , [%], [%] |
|---|---|---|---|---|
| 1000 | 4.29, 1.38, 0.957, 1.85, 0.13 | 4.47, 0.86, 0.970, 1.85, 0.12 | 3.35, 0.83, 0.982, 1.83, 0.13 | 3.46, 0.95, 0.975, 1.84, 0.13 |
| 4MWs | 4.28, 1.36, 0.959, 1.79, 0.14 | 4.49, 0.85, 0.971, 1.78, 0.14 | 3.34, 0.82, 0.982, 1.77, 0.16 | 3.45, 0.93, 0.976, 1.77, 0.15 |
| 5MWs | 4.35, 1.32, 0.962, 1.68, 0.15 | 4.53, 0.82, 0.973, 1.66, 0.15 | 3.41, 0.81, 0.983, 1.65, 0.15 | 3.50, 0.91, 0.977, 1.65, 0.15 |
| 1000T | 4.83, 1.97, 0.926, 0.52, 0.13 | 3.79, 0.82, 0.972, 0.51, 0.02 | 3.04, 0.73, 0.986, 0.51, 0.03 | 3.12, 0.90, 0.977, 0.51, 0.04 |
| 4MWsT | 4.84, 1.90, 0.934, 0.44, 0.10 | 3.97, 0.66, 0.981, 0.42, 0.03 | 3.32, 0.68, 0.988, 0.42, 0.03 | 3.40, 0.83, 0.981, 0.42, 0.04 |
| 5MWsT | 4.81, 1.82, 0.940, 0.40, 0.07 | 4.15, 0.63, 0.983, 0.39, 0.03 | 3.44, 0.67, 0.988, 0.39, 0.04 | 3.53, 0.81, 0.982, 0.39, 0.04 |

Table 2. Summary of statistics for FTIR-Brewer comparison for the set-ups 1000/1000T, 4MWs/4MWsT, and 5MWs/5MWsT: median (M, in %) and standard deviation ($\sigma$, in %) of the relative differences, and Pearson correlation coefficient (R) of the direct comparison for the periods 1999-2004, 2005-May 2008 and June 2008-2018, and for the entire time series (1999-2018). Also the median and standard deviation of the theoretical total statistical errors for the different FTIR set-ups and periods.

*- p. 15, l. 335: "inconsistency in the parametrisation of the spectroscopic parameters at higher wavenumbers"*

*Do you mean that at 1012 cm-1 the spectroscopic parameters linked to temperature dependence are not consistent? Did you check the origin (studies) used for the parameters in hitran? Is it different studies for 1000-1005 cm-1 and 1012 cm-1?*

Actually, the number of O₃ observations for slant columns greater than 800 DU recorded at IZO does not allow robust conclusions to be obtained. In this sense, including NDACC FTIR stations at higher latitudes would help us to analyse whether there is indeed a different behaviour depending on the O₃slant column, as suggested from Figure 7 of the preprint.

What is evident from Figure 7 is that the O₃ retrievals from the 1000T set-up significantly differ from those obtained using the narrow micro-windows. As a possible explanation, the authors point to possible inconsistencies in the spectroscopic parameters, since an erroneous parameterisation of the temperature dependence of the O₃ line width may produces systematic differences between actual and retrieved temperature profiles (Schneider and Hase, 2008), therefore affecting the absolute value of O₃ FTIR products. Note that both 4MWsT and 5MWsT set-ups behave consistently. Consequently, if such inconsistency exists, it would affect the common O₃ lines of both 4MWsT and 5MWsT set-ups.

The authors have carefully looked for some reference to that issue in the literature, especially in the reference papers of HITRAN spectroscopic database (used in the current work) (Rothman et al., 2005, 2009, 2019; Gordon et al., 2017). The major improvements with respect to the 1000 cm⁻¹ O₃ band were carried out in the updates of HITRAN 2004 and HITRAN 2016 databases;

however detailed information about specific absorbing lines was not found. In order to analyse the impact of the different versions of spectroscopic database on the FTIR $O_3$ retrievals, Figure 6 shows the differences between the retrieved $O_3$ total columns from the different set-ups using the 2004, 2012 and 2016 versions of the HITRAN spectroscopic database with respect to the 2008 version, which was applied in the current work following the NDACC IRWG guidelines, as well as the total systematic errors for the $O_3$ total columns for all the set-ups considered and different HITRAN versions. As mentioned before, the major differences are found for the 2004 and 2016 versions, reaching differences between 0.2% and -0.3% when using HITRAN 2016. It is worth highlighting the strong impact of the temperature retrieval on the retrieved $O_3$ total columns, changing the sign of differences for both HITRAN 2004 and 2016 versions. The findings further confirm the important impact of spectroscopic line parametrization of FTIR retrievals and motivate detailed analyses when the new version of HITRAN spectroscopic database (HITRAN 2020) is released in the coming months (see details at www.hitran.org).

[Figure]

Figure 6. (a) Differences between the retrieved $O_3$ total columns from the different set-ups using the 2004, 2012 and 2016 versions of the HITRAN spectroscopic database with respect to the 2008 version, which was applied in the current work, as a function of the $O_3$ slant columns [DU] for the exemplary day used in the paper (31st August 2007). (b) Total systematic error for the $O_3$ total columns for all the set-ups considered and different HITRAN versions. Note that the assumed error value for the spectroscopic parameters was 3%, according to the revised uncertainty estimation.

Another point distinguishing largely the broad and narrow set-ups is the potential $H_2O$ interference due to the presence of strong $H_2O$ absorbing lines in the 1000 $cm^{-1}$ region (recall Figure 1). Although the two-step strategy minimises the $H_2O$ effect, as observed in Figure 2, the $H_2O$ interference is expected to increase as air mass increases (higher $O_3$ slant columns) and to be especially remarkable for the 1000T set-up.

This explanation will be included in Section 4.1 of the revised manuscript.

*- p.17, L. 351: "the scatter found is noticeably lower than that predicted when the temperature fit is not considered"*

*Indeed. This would mean that the a-priori temperature covariance matrix (SaT), constructed following Schneider et al. (2008a), is chosen with too large uncertainty parameters (-3.5K at the surface to + 4K at 30km). This is quite an important statement since the theoretical demonstration that the temperature fit is improve the retrievals (when stable instrument) is based on this SaT matrix (which presently gives large theoretical uncertainty when T is not retrieved).*

*This should be recalled also in the conclusions, p. 23, l. 445: "Theoretically, the total error of O3 TCs is halved when applying a temperature fit": probably the effect will be less if smaller values in SaT are used (as suggested by the observed scatter).*

Indeed, Schneider et al. (2008) and Schneider and Hase (2008) presented for the first time the simultaneous optimal estimation of $O_3$ and temperature profiles from measured FTIR spectra, using the same temperature information for the a-priori temperature covariance matrix and the error temperature matrix as stated by the Referee. However, in the current work, the assumed temperature uncertainties were reduced with respect to Schneider's works to better fit with the expected uncertainty of the NCEP profiles (e.g. Langland et al., 2008), according to previous works (e.g. García et al., 2012; García et al., 2014a; Sepúlveda et al., 2014). As included in Appendix A of the preprint, the temperature error was assumed to be 2K and 5K below and above 50 km of altitude, respectively. 70% of these errors can be ascribed to the statistical contribution while the remaining 30% to the systematic contribution.

[Figure]

Figure 7. Estimated total statistical (ST) and systematic (SY) errors (in %) for O3 TCs retrieved from the set-ups 1000/1000T (a and b), and 5MWs/5MWsT (c and d) as a function of $O_3$ slant column (in DU) for the FTIR measurements taken on 31st August 2007 from SZAs between 84º (~07:00 UT) and 21º (~13:30 UT). (e), (f), (g), (h) as for (a), (b), (c), and (d), but for different atmospheric temperature error profiles.

The authors agree with the Referee in that the theoretical estimation of the temperature fit retrieval is a key issue in the current work as it strongly affects the total uncertainty budget presented. Therefore, in order to account for different casuistry, the uncertainty assessment for different sets of atmospheric temperature error profiles will be included in Appendix A, similarly to the ILS analysis, and briefly discussed in Section 3.2.2 and Conclusions. Accordingly, Figure A1 of the preprint will be replaced by Figure 7 above, which shows how the statistical error fits better with the scatter observed between FTIR and Brewer observations when smaller uncertainties are assumed for the temperature contribution.

*- p. 17: discussion seasonal cycle l. 355-365:*

*I suggest to add scatter plots Brewer vs FTIR set-ups in Fig. 8. Offset and slope will distinguish the constant bias between Brewer and FTIR and the proportional one (which gives a seasonal effect of RD)*

Figure 8 will be redone including the scatter plots of Brewer versus FTIR set-ups suggested by the Referee.

*- p. 18; l. 381 and p.20 Table 3: comparison at the representative altitudes of 5, 18, and 29 km:*

*Why not using partial columns comparisons as in García et al. (2012)? It should be more stable because the wider layers are then less dependent on the smoothing error, less dependent on the DOFS (which are quite variable, especially in the 120M period, Table 1) than a single point on the profile. Overall smaller uncertainty on wider layers than on single point profile.*

*This "single point" comparisons of scatter vs theoretical error budget (cf. the discussion p. 21 l.407) is also then not straightforward because the uncertainty profiles (Fig.5, Sect3.2.2.) are not independent (the covariance matrices are not diagonal).*

Following the Referre's suggestion, the FTIR and ECC comparison at representative altitudes will be replaced by the comparison between ozone partial columns using the altitude levels as defined in García et al. (2012), i.e., the layers that are sufficiently well-detectable by the ground-based FTIR system (2.37-13 km, 12.-23 km and 22-29 km). Therefore, the following Table will replace Table 3 of the preprint and the discussion of Section 4.2 will be modified accordingly.

| set-up | 1999-2004 M[%], $\sigma$[%], R | 2005-2008 M[%], $\sigma$[%], R | 2008-2018 M[%], $\sigma$[%], R | 1999-2018 M[%], $\sigma$[%], R |
|---|---|---|---|---|
| | **FTIR-ECC at 2.37-13 km** | | | |
| 1000 | 15.86, 7.08, 0.934 | 11.44, 6.58, 0.956 | 9.08, 5.46, 0.970 | 10.63, 6.61, 0.953 |
| 4MWs | 15.08, 7.11, 0.937 | 12.08, 6.54, 0.958 | 9.38, 5.10, 0.974 | 10.62, 6.46, 0.956 |
| 5MWs | 16.24, 7.09, 0.938 | 12.30, 6.64, 0.957 | 9.72, 5.13, 0.974 | 10.89, 6.56, 0.956 |
| 1000T | 15.05, 7.30, 0.934 | 10.39, 6.73, 0.954 | 8.12, 5.45, 0.971 | 10.00, 6.67, 0.953 |
| 4MWsT | 15.56, 7.22, 0.938 | 11.56, 6.76, 0.957 | 9.26, 4.94, 0.975 | 10.54, 6.36, 0.958 |
| 5MWsT | 16.93, 7.05, 0.940 | 12.76, 6.82, 0.956 | 10.11, 4.99, 0.976 | 11.20, 6.42, 0.958 |
| | **FTIR-ECC at 12-23 km** | | | |
| 1000 | 17.30, 5.76, 0.914 | 16.01, 4.54, 0.946 | 13.23, 4.70, 0.959 | 14.59, 5.23, 0.944 |
| 4MWs | 17.91, 5.58, 0.928 | 15.99, 4.86, 0.943 | 13.41, 4.82, 0.961 | 14.79, 5.29, 0.947 |
| 5MWs | 17.89, 5.43, 0.939 | 16.03, 4.94, 0.943 | 13.72, 4.81, 0.962 | 14.91, 5.21, 0.951 |
| 1000T | 16.83, 5.64, 0.921 | 15.19, 4.59, 0.945 | 12.13, 4.75, 0.961 | 13.72, 5.27, 0.946 |
| 4MWsT | 17.25, 5.61, 0.930 | 16.44, 4.65, 0.947 | 14.18, 4.73, 0.963 | 15.30, 5.12, 0.951 |
| 5MWsT | 17.48, 5.72, 0.935 | 16.91, 4.67, 0.948 | 15.07, 4.77, 0.963 | 16.02, 5.11, 0.953 |
| | **FTIR-ECC at 22-29 km** | | | |
| 1000 | 15.92, 3.71, 0.820 | 16.23, 2.87, 0.888 | 13.94, 3.17, 0.756 | 14.76, 3.50, 0.779 |
| 4MWs | 16.09, 4.01, 0.793 | 16.67, 2.88, 0.888 | 14.16, 3.21, 0.767 | 14.80, 3.64, 0.773 |
| 5MWs | 16.00, 4.10, 0.783 | 16.43, 2.87, 0.892 | 14.06, 3.23, 0.777 | 14.79, 3.68, 0.776 |
| 1000T | 16.80, 3.54, 0.866 | 15.56, 3.12, 0.856 | 13.88, 3.22, 0.735 | 14.51, 3.55, 0.778 |
| 4MWsT | 16.28, 3.56, 0.864 | 15.61, 3.36, 0.844 | 14.45, 3.05, 0.774 | 14.96, 3.43, 0.801 |
| 5MWsT | 16.31, 3.67, 0.856 | 15.28, 3.53, 0.829 | 14.47, 3.12, 0.775 | 15.06, 3.50, 0.798 |

Table 3: Summary of statistics for the FTIR-smoothed ECC comparison for the $O_3$ partial columns computed between 2.37-13 km, 12-23 km, and 22-29 km for the set-ups 1000/1000T, 4MWs/4MWsT, and 5MWs/5MWsT: median (M, in %) and standard deviation ($\sigma$, in %) of the relative differences, and Pearson correlation coefficient (R) of the direct comparison for the periods 1999-2004, 2005-May 2008 and June 2008-2018, and for the entire time series (1999-2018). The number of coincident FTIR-ECC measurements is 56, 49, and 167 for the three periods, respectively, and 272 for the whole dataset.

*- p.23, l.437: "Quality of the FTIR O3 products improves as the retrieval strategies become more refined by including O3 absorption lines in specific narrow micro-window"*

*The conclusions are less clear when comparing to sondes (p. 21, l. 420), and this should also be written in the conclusions. Probably a clearer conclusion would have helped to convince the IRWG (more than 20 sites) to re-perform their retrievals, re-archive in NDACC their data, using improved settings. The very detailed and careful analysis performed in this study could have had more fast and clear impact on the IRWG if it would have been applied to at least 1 or 2 other sites. This could have helped to strengthen the findings (humid sites for the effect of narrow mws avoiding strong H2O lines present in the broad mw; site with coincident Lidar measurements to check the effect of retrieval settings at higher altitudes, where the expected ozone recovery*

*should be detected first). Let's hope volunteer sites will try this exercise now independently, otherwise the impact of the current study on the IRWG harmonization will be more limited.*

Effectively, the vertical comparison to reference ozonosondes is less conclusive than for total columns. In this sense, the authors have sought to present and discuss the results obtained, but we do not find it appropriate to recommend different set-ups depending on altitude ranges as this recommendation could be confusing and not practical for operational retrievals within the NDACC FTIR community. This will be included in the Conclusions as requested by the Referee.

As mentioned before, the authors fully agree with the Referee in that testing the proposed O3 set-ups on different NDACC FTIR stations (under different humidity conditions, latitudes, altitudes...) would indeed motivate the NADCC FTIR community to revise the standard O3 retrieval strategy. In this sense, discussions have already started with other NDACC FTIR stations to carry out a harmonised testing. However, the authors consider that such huge exercise needs to be carefully planned and carried out to account for the existing casuistry among the NDACC FTIR sites in another separated work. In order to strengthen the results obtained at IZO and make them more representative of different environments, significant information will be added to the original manuscript (e.g. assessment of water vapour interference and discussion of more detailed uncertainty budget, including the smoothing error or sensitivity on temperature error profiles).

*Minor or technical comments:*

*- Section Introduction, l.26 and following places: "O3 measurements...." Specify that you are talking about o3 total and/or stratospheric ozone measurements when you discuss ozone decline.*

*- p. 9, l .221: "do depend" instead of "do dependent"?*

*- p. 10, Fig. 4: "for O3 TCs...": in the text you use "O3 SC". Same remark for p.11 l. 224 & 258: SC instead of TC?*

*- p. 11. L. 243 and l. 254: 1000T (not 100T)*

*- p. 14, Figure 6: explains what represents the shaded areas (1-sigma, 2-sigma of the monthly means?). Maybe enlarge / add some grids, to better see steps / better compare (b) and (c).*

All technical comments have been corrected in the revised manuscript according to the Referee's suggestions.

**References**

Birk, M., G. Wagner, I. E. Gordon, B. J. Drouin, Ozone intensities in the rotational bands, Journal of Quantitative Spectroscopy and Radiative Transfer, 226, 60-65, ISSN 0022-4073, https://doi.org/10.1016/j.jqsrt.2019.01.004, 2019.

Drouin, B.J., T. J. Crawford, S. Yu, Validation of ozone intensities at 10 μm with THz spectrometry, Journal of Quantitative Spectroscopy and Radiative Transfer, 203, 282-292, ISSN 0022-4073, https://doi.org/10.1016/j.jqsrt.2017.06.035, 2017.

García, O. E., Schneider, M., Redondas, A., González, Y., Hase, F., Blumenstock, T., and Sepúlveda, E.: Investigating the long-term evolution of subtropical ozone profiles applying ground-based FTIR spectrometry, Atmospheric Measurement Techniques, 5, 2917–2931, https://doi.org/10.5194/amt-5-2917-2012, 2012.

García, O. E., Schneider, M., Hase, F., Blumenstock, T., Sepúlveda, E., and González, Y.: Quality assessment of ozone total column amounts as monitored by ground-based solar absorption spectrometry in the near infrared (>3000 cm−1), Atmospheric Measurement Techniques, 7, 3071–3084, https://doi.org/10.5194/amt-7-3071-2014, 2014a.

García, O.E., M. Schneider, F. Hase, T. Blumenstock, E. Sepúlveda, E. Cuevas, A. Redondas, A. Gómez-Peláez and J.J. Bustos, Effect of updates in LINEFTIT and PROFFIT at Izaña ground-based FTIR: improvement of the ILS characterisation and intra-day pressure and temperature variability, NDACC-IRWG/TCCON meeting 2014, 12-16 May, Bad Sulza (Germany), 2014b.

García, O., Sanromá, E., Schneider, M., Hase, F., León-Luis, S.F., Blumenstock, T., Sepúlveda, E., Redondas, A., Carreño, V., Torres, C., and Prats, N., Ozone monitoring by NDACC FTIR spectrometry: improved retrieval strategy and impact of instrumental line shape characterization, TCCON/COCCON/NDACC Meeting 2021 (online), 8-10 June, 2021.

Gordon, I.E., L.S. Rothman, C. Hill et al., The HITRAN2016 Molecular Spectroscopic Database, Journal of Quantitative Spectroscopy and Radiative Transfer 203, 3-69, 2017.

Gröbner, J., Redondas, A., Weber, M., and Bais, A.: Final report of the project Traceability for atmospheric total column ozone (ENV59, ATMOZ), Tech. rep., EURAMET, https://www.euramet.org/research-innovation/search-research-projects/details/project/traceability-for-atmospheric-total-column-ozone/, 2017.

Hargreaves, R., I. Gordon, L. Rothman, R. Hashemi, E. Karlovets, F. Skinner, E. Conway, Y. Tan, C. Hill, and R. Kochanov, HITRAN2020: An overview of what to expect, EGU General Assembly, Session AS5.12 (D3293|EGU2020-20533), 5 May, 2020.

IRWG: Uniform Retrieval Parameter Summary, http://www.acom.ucar.edu/irwg/IRWG_Uniform_RP_Summary-3.pdf, 2014.

Rolf H. Langland, R. H., R. N. Maue and C. H. Bishop, Uncertainty in atmospheric temperature analyses, Tellus A: Dynamic Meteorology and Oceanography, 60:4, 598-603, DOI: 10.1111/j.1600-0870.2007.00336.x, 2008.

Rothman, L.S., I. E. Gordon, Y. Babikov et al., The HITRAN 2012 Molecular Spectroscopic Database, Journal of Quantitative Spectroscopy and Radiative Transfer 130, 4-50 2013.

Rothman, L.S., I. E. Gordon, A. Barbe et al., The HITRAN 2008 Molecular Spectroscopic Database, Journal Of Quantitative Spectroscopy and Radiative Transfer 110, 533-572, 2009.

Rothman, L.S., D. Jacquemart, A. Barbe et al., The HITRAN 2004 molecular spectroscopic database, Journal of Quantitative Spectroscopy and Radiative Transfer 96, 139-204, 2005.

Schneider, M. and Hase, F.: Technical Note: Recipe for monitoring of total ozone with a precision of around 1 DU applying mid-infrared solar absorption spectra, Atmospheric Chemistry and Physics, 8, 63–71, https://doi.org/10.5194/acp-8-63-2008, 2008.

Schneider, M., Hase, F., Blumenstock, T., Redondas, A., and Cuevas, E.: Quality assessment of O3 profiles measured by a state-of-the-art ground-based FTIR observing system, Atmospheric Chemistry and Physics, 8, 5579–5588, 2008b.

Schneider, M., Barthlott, S., Hase, F., González, Y., Yoshimura, K., García, O. E., Sepúlveda, E., Gomez-Pelaez, A., Gisi, M., Kohlhepp, R., Dohe, S., Blumenstock, T., Wiegele, A., Christner, E., Strong, K., Weaver, D., Palm, M., Deutscher, N. M., Warneke, T., Notholt, J., Lejeune, B., Demoulin, P., Jones, N., Griffith, D. W. T., Smale, D., and Robinson, J.: Ground-based remote sensing of tropospheric water vapour isotopologues within the project MUSICA, Atmospheric Measurement Techniques, 5, 3007–3027, https://doi.org/10.5194/amt-5- 3007-2012, 2012.

Sepúlveda, E., Schneider, M., Hase, F., Barthlott, S., Dubravica, D., García, O. E., Gomez-Pelaez, A., González, Y., Guerra, J. C., Gisi, M., Kohlhepp, R., Dohe, S., Blumenstock, T., Strong, K., Weaver, D., Palm, M., Sadeghi, A., Deutscher, N. M., Warneke, T., Notholt, J., Jones, N., Griffith, D. W. T., Smale, D., Brailsford, G. W., Robinson, J., Meinhardt, F., Steinbacher, M., Aalto, T., and Worthy, D.: Tropospheric CH4 signals as observed by NDACC FTIR at globally distributed sites and comparison to GAW surface in situ measurements, Atmos. Meas. Tech., 7, 2337–2360, https://doi.org/10.5194/amt-7-2337-2014, 2014.

Sun, Y., Palm, M., Liu, C., Hase, F., Griffith, D., Weinzierl, C., Petri, C., Wang, W., and Notholt, J.: The influence of instrumental line shape degradation on NDACC gas retrievals: total column and profile, Atmospheric Measurement Techniques, 11, 2879–2896, https://doi.org/10.5194/amt-11-2879-2018, 2018.

Takele Kenea, S., Mengistu Tsidu, G., Blumenstock, T., Hase, F., von Clarmann, T., and Stiller, G. P.: Retrieval and satellite intercomparison of O3 measurements from ground-based FTIR Spectrometer at Equatorial Station: Addis Ababa, Ethiopia, Atmospheric Measurement Techniques, 6, 495–509, https://doi.org/10.5194/amt-6-495-2013, 2013.

Tyuterev, VI. G., A. Barbe, D. Jacquemart, C. Janssen, S. N. Mikhailenko, and E. N. Starikova, Ab initio predictions and laboratory validation for consistent ozone intensities in the MW, 10 and 5 $\mu$μm ranges, J. Chem. Phys. 150, 184303, https://doi.org/10.1063/1.5089134, 2019.

Vigouroux, C., de Mazière, M., Demoulin, P., Servais, C., Hase, F., Blumenstock, T., Kramer, I., Schneider, M., Mellqvist, J., Strandberg, A., Velazco, V., Notholt, J., Sussmann, R., Stremme, W., Rockmann, A., Gardiner, T., Coleman, M., and Woods, P.: Evaluation of tropospheric and stratospheric ozone trends over Western Europe from ground-based FTIR network observations, Atmospheric Chemistry and Physics, 8, 6865–6886, https://doi.org/10.5194/acp-8-6865-2008, 2008.

Vigouroux, C., Blumenstock, T., Coffey, M., Errera, Q., García, O., Jones, N. B., Hannigan, J. W., Hase, F., Liley, B., Mahieu, E., Mellqvist, J., Notholt, J., Palm, M., Persson, G., Schneider, M., Servais, C., Smale, D., Thölix, L., and De Mazière, M.: Trends of ozone total columns and vertical distribution from FTIR observations at eight NDACC stations around the globe, Atmospheric Chemistry and Physics, 15, 2915–2933, https://doi.org/10.5194/acp-15-2915-2015, 2015.

WMO: Quality assurance and quality control for ozonesonde measurements in GAW, World Meteorological Organization (WMO)-Report No. 201, Tech. rep., Smit, H.G.J., and ASOPOS panel (eds.), World Meteorological Organization, Geneva, Switzerland, 2014.

---

## Author Response (AR1)

Dear Editor,

First, I would like to thank the Editorial team for the extension of the revision phase of this manuscript. The volcano the eruption on the island of La Palma was an extra (and hard) workload for most of the co-authors of this manuscript last year. Therefore, we had to postpone temporally this work.

All comments and suggestions from both Referees have been assessed and included in the revised manuscript.

Many thanks and best regards,

Omaira García et al.

---

## Author Response (AR2)

Dear Editor,

All comments suggestions from you and Referee #1 have been assessed and included in the revised manuscript.

Please find below the response to the Referee #1's comments (in bold her comments and italic the authors' replies).

Many thanks and best regards,

Omaira García et al.

**Response to Referee#1**

**Minor comments**

**I would like to thank the authors for the additional work they performed for this updated manuscript, especially concerning the Appendix A on the uncertainties linked to the H2O interferences. However, I would conclude differently (or at least be more nuanced) in the Appendix A (and therefore in the summary and Conclusions section). Indeed, looking at Fig A1, we see: - The theoretical H2O uncertainties in the range of slant column of IZO (so in the range where we can verify it), are very small and similar for one-step / two-step (when no temperature retrieval). And even (unexpectedly?) smaller with 1000 set-up in the one-step approach. (when temperature is retrieved, we can question the theoretical results since they give higher theoretical H2O error, while the empirical ones are smaller.) - The empirical verification (comparisons with Brewer) shows no improvement at all (even worse for the 1000T) when using the two-step approach, when the instrument is stable (2008-20018). And the improvement for the period 2005-2008 is small (0.01-0.02%). So, I would conclude from Appendix A that both approaches are valid to correctly take into account the H2O interferences.**

*The author agree with the Referee and conclusions regarding the one- and two-steps retrieval strategies have been modified following the Referee's suggestions as follows:*

*In Appendix A: "One-step and two-step approaches provide consistent results when the simultaneous temperature fit is not included for all set-ups and, therefore, both can be valid to correctly minimize the H2O interference. Nonetheless, provided the documented improvement of the temperature retrieval is pursued, the two-step strategy ought to be used. In this sense, the two-step strategy drastically reduces the H2O interfering error for those set-ups using narrow micro-windows when the simultaneous temperature fit is included (4MWsT/5MWsT), leading to expected errors on the O3 total columns smaller than 0.01%. The H2O interfering effect also drops for the 1000 spectral region, but to a lesser extent, given the presence of important H2O absorption lines in that region (recall Figure 1). This should be especially taken into account for FTIR stations located in humid environments.*

*The comparison to Brewer observations (Figure A1 (c) and (d)) also corroborates the theoretical results. It is worth highlighting the fact that the differences found between the two strategies are in excellent agreement with the estimated H2O interfering error values (Figure A1 (a) and (b))."*

*In Conclusions: "In this sense, using one-step or two-step retrieval strategies (retrieving H2O and O3 in the same or in two separated steps, respectively) has been found to be valid and provide consistent results."*

**Another remark for the Summary-Conclusion Section: There is indeed no doubt that the narrow micro-windows (and T retrieval with stable instrument) set-up has been proven better by the authors, in terms of O3 total column precision,. But it's worth mentioning in the conclusions (as it has been said in the text), that this is not the case for the O3 profiles' precision, depending on the altitude range that is considered.**

*The following statement has been included in the Conclusion:*

*Regarding the vertical O3 distribution, the important cross-interference between the O3 and temperature profiles, and the instrumental status results in a differentiated performance of the set-ups depending on the altitude range. The best overall performance is documented for the set-ups using narrow micro-windows and simultaneous temperature fits in the troposphere and stratosphere regions, while in the tropopause altitudes the broad micro-window strategy seems to provide the best agreement with respect to ozonosonde data.*

**Technical comments:**

**- Figure 3 b) and c): I do not see the TE 1000, TE 4MWs, and TE 5MWS lines: is it because they are covered by other lines or are they missing?**

*Indeed these TE errors are included in Figure 3, but they are covered by the other lines (they can be slightly distinguished at low ozone slant columns).*

**- P. 14, l. 348: "Including o not this fit…": the "r" is missing**

Corrected

**- P. 15, l. 355-356: "and instrument status": we do not understand this conclusion here. I guess this is because in the AMTD version, the authors said that the situation is different with bruker HR vs Bruker M, but here it has been removed. So ?**

*The Referee is right. This statement has been modified as follows: "This result further corroborates that the broad region seems to be less sensitive to the improvement generated by the temperature retrieval."*

**- P. 21 l. 469: NDACC not NADCC**

Corrected